



# Predictive Performances of Machine Learning– and Deep Learning–Based Univariate and Multivariate Reservoir Inflow Predictions in the Chao Phraya River Basin

Jidapa Kraisangka[a], Pheeranat Dornpunya[b, c], Areeya Rittima[b*], Wudhichart Sawangphol[a], Yutthana Phankamolsil[d], Allan Sriratana Tabucanon[e], Yutthana Talaluxmana[f], and Varawoot Vudhivanich[g]

[a] Faculty of Information and Communication Technology, Mahidol University, Thailand
[b, b*] Graduate Program in Environmental and Water Resources Engineering, Department of Civil and Environmental Engineering, Faculty of Engineering, Mahidol University, Thailand
[c] Hydro–Informatics Institute (Public Organization), Thailand
[d] Environmental Engineering and Disaster Management Program, Mahidol University, Kanchanaburi Campus, Thailand
[e] Faculty of Environment and Resource Studies, Mahidol University, Phuttamonthon, Nakhon Pathom, Thailand
[f] Department of Water Resources Engineering, Faculty of Engineering, Kasetsart University, Thailand
[g] Department of Irrigation Engineering, Faculty of Engineering at Kamphaeng Saen, Kasetsart University, Thailand

*Corresponding author(s). E–mail(s): areeya.rit@mahidol.ac.th; Contributing authors: jidapa.kra@mahidol.ac.th; pheeranat.dor@gmail.com; wudhichart.saw@mahidol.ac.th; yutthana.pha@mahidol.ac.th; allansriratana.tab@mahidol.ac.th; fengynt@ku.ac.th; fengvwv@ku.ac.th

**Abstract**

This study demonstrated the predictability of Machine Learning (ML)– and Deep Learning (DL)–based univariate and multivariate predictions of reservoir inflows of Bhumibol (BB) and Sirikit (SK), two major dams in the Chao Phraya River Basin. XGBoost, tree–based ensemble–, and LSTM, deep neural network–based algorithms were selected for development of daily and monthly prediction models. For univariate prediction, the inflows of the BB and SK dams were predicted separately using two individual models. In contrast, for multivariate prediction, a single model was developed to simultaneously predict the inflows of both the BB and SK dams facilitating the integrated decision–making processes. Across all prediction scenarios, ML– and DL–based models demonstrated superior performances in predicting daily reservoir inflows for BB and SK dams compared to monthly predictions, achieving NSE values of 0.86 and 0.77, respectively. Since modeling with LSTM algorithm can effectively handle larger datasets, this enables single multivariate prediction model to predict closer results to those individual univariate models performed by XGBoost and LSTM for BB and SK prediction. XGBoost models mostly outperformed LSTM when tested on the datasets for both daily and monthly univariate predictions. Among all prediction scenarios, underprediction of low reservoir inflows and overprediction of high reservoir inflows by both univariate and multivariate models were consistently existed. Therefore, extracting specific and informative insights from the results of each model type, forecasting horizon, and algorithms used can significantly enhance decision–making support for both real–time reservoir operation and long–term reservoir management planning.

**Keywords:** Machine Learning (ML), Deep Learning (DL), Artificial Intelligence (AI), Reservoir Inflow Prediction, Chao Phraya River Basin

## 1. Introduction

The increased climate variability has intensified the water–related challenges globally making the water resources management more complicated under the changing circumstances (Ngamsanroaj and Tamee, 2019). Consequently, risks of water stress and scarcity exacerbated by reservoir operation and management have substantially increased. A key factor contributing to this is uncertain water supply in the reservoir system. Reservoir inflow is commonly considered as the principal source of water supply in reservoirs. It is significantly influenced by the climate variability and hydrologic phenomena. Accurate and precise reservoir inflow prediction plays a crucial role for effective reservoir planning and management (Suprayogi et al., 2020). During critical climate events, future reservoir inflow forecast informs decision making for instant operational responses to natural disasters like floods and drought. During normal climate conditions, predictive reservoir data is used for establishment of guideline trajectory for proper reservoir operation. Predicting the precise reservoir inflow is inherently stochastic due to uncertainty of hydrological inputs and strong non–linearity of the system (Soncin et al., 2024). For decades, numerous research studies have been dedicated to reservoir inflow predictions to achieve the desired purposes for both reservoir management planning and real–time operation.

A wide range of prediction techniques including physically process–based–, conventional stochastic–based– and modern data–driven approaches like Machine Learning (ML) and hybrid models have been widely employed to enhance the model predictability. It is revealed that the process–based hydrological models usually rely on a various number of assumptions and require many physical parameters to resemble the hydrological nature of environment (Firat and Güngör, 2008; Luo et al., 2020). In addition, the conventional stochastic–based techniques such as ARMA, ARIMA for time series prediction provided good results for only linear data. However, it is not appropriate for non–



linearity phenomenon influenced by climate, natural geography, and human activities (Valipour et al., 2013).
Prediction time horizon selected has been generally ranged from short–term (hourly, daily, weekly) to long–term
predictions (monthly, seasonal, yearly). Univariate prediction model is generally used to predict future values of single
variable using the historical data. While, multivariate prediction model involves predicting the future values of
multiple interrelated variables. However, short–term univariate prediction predicting future values of single reservoir
inflow has broadly been found particularly for real–time operation and short–term planning. Predicting multiple
reservoir inflows using multivariate prediction model by taking the interdependencies of influencing factors of
multiple reservoirs has rarely studied and limited.
The advancement of Artificial Intelligence (AI) and Machine Learning (ML) approach has revolutionized
transformative impacts across the traditional prediction methods and their disciplines. A great deal of data–driven ML
approach has enhanced the predictability for hydrological prediction (Zhang et al., 2018) such as rainfall (Chen et al.,
2017; Ridwan et al., 2021), streamflow (Latif et al., 2023; Kisi et al., 2024), reservoir inflow (Zhang et al., 2021;
Hameed et al., 2022; Latif et al., 2024), reservoir water level (Sapitang et al., 2020; Aquil and Ishak, 2023), river water
level (Ahmed et al., 2023; Zakaria et al., 2023), groundwater level (Osman et al., 2020), sediment transport
(Almubaidin et al., 2023), water quality prediction (Haghiabi et al., 2018; Shams et al., 2024) and snow water
equivalent (Khosravi et al., 2023).
Since 1990s, a well–known Artificial Neural Networks (ANNs) inspired by human brain structure and its
function has intensively been used for hydrological prediction. It was commonly applied for both univariate and
multivariate predictions. For example, univariate time series prediction of reservoir inflow was developed using ANNs
to map the non–linear relationships between input and output variables (Kawade et al., 2019). Many studies also
revealed that ANNs significantly outperformed than the statistically–stochastic–based prediction models like AR, MA,
and ARIMA for the reservoir inflow prediction (Pradeepakumari and Srinivasu, 2019). Additionally, ANNs can be
applicable for both parametric and non–parameter data (Pini et al., 2020).
The rapid evolution of ML algorithms has been driven by the advancement and succession of computer
science and technologies to increase the computational capability and handle large dataset. Consequently, the
improved ML algorithms have been progressively developed incorporating supervised learning, unsupervised
learning, and reinforcement learning for various tasks. Several conventional ML algorithms have commonly been
employed for reservoir inflow prediction such as Support Vector Machines (SVM), K–Nearest Neighbors (KNN),
Random Forest (RF), Multi–layer Perceptron (MLP), Gradient Boosting (GB), Extreme Gradient Boosting
(XGBoost), and Radial Bias Function (RBF). A comparative study for daily reservoir inflow prediction was conducted
using four different approaches; (1) Multiple Linear Regression (MLR), (2) Random Forest (RF), (3) Extreme
Learning Machine (ELM), and (4) Regularized Extreme Learning Machine (RELM). The results showed the
superiority of RELM approach that yielded higher prediction accuracy with R = 0.955 (Hameed et al., 2022).
XGBoost, a relatively recent algorithm, has proven its effectiveness in reservoir inflow prediction. To forecast multi–
step ahead daily and monthly reservoir inflows, XGBoost was ranked as the best prediction model compared to MLP,
Support Vector Regression (SVR), Adaptive Neuro–Fuzzy Inference System (ANFIS) (Ibrahim et al., 2023). Similarly,
XGBoost outperformed RF and an ensemble model combining XGBoost–RF algorithms for daily reservoir inflow
prediction (Jan et al., 2024)
Recently, achievement of Deep Learning (DL) which is a subset of ML, has renowned incorporating multiple
layers artificial neural networks or Deep Neural Networks (DNNs) with automatic feature learning. It is indicated that
DL is powerful in extracting complex features hidden in vast amount of hydrological dataset (Huang et al., 2022).
Some of the most widely used DL algorithms for time series prediction are Convolutional Neural Networks (CNNs),
Recurrent Neural Networks (RNNs), Long–Short Term Memory (LSTM), and Gated Recurrent Unit (GRU). GRU is
the simplified form of LSTM well–suited for faster training for time series prediction. While, LSTM is a specific type
of RNNs designed to learn the long–term temporal dependency and seasonality of time series data and overcome the
vanishing and exploding gradient problems of RNNs (Dtissibe et al., 2024). It is found that RNNs with input delayed
time gave better predictive performances for multivariate reservoir inflow prediction than Input Delayed Neural
Network (IDNN) (Coulibaly et al., 2001). Real–time reservoir inflow prediction in the case of different climate
scenarios and lead time conditions (+1–Hr, +4–Hr, and +6–Hr) with three conventional ML algorithms; (1) SVM, (2)
RF, (3) MLP and four DL algorithms; (1) DNNs, (2) RNNs, (3) LSTM, (4) GRU were investigated. In comparation,
the results distinctly showed that DNNs outperformed than the conventional ANNs in most scenarios. However, under
the extended periods of lead time prediction, underestimation of reservoir inflow by DDNs is more serious compared
to ANNs (Huang et al., 2022). Encoder–Decoder LSTM (ED–LSTM) was employed for sub–seasonal reservoir inflow
with multiple lead time (+1–D to +30–D) for 30 reservoirs, at 1–D ahead prediction, ED–LSTM could produce good
predictive performance of NSE exceeding 0.75 for 29 reservoirs. At 30–D ahead prediction, ED–LSTM achieved NSE
of more than 0.5 for most reservoirs (Fan et al., 2023). Furthermore, LSTM exhibited superior performances in





predicting medium to long–term data compared to the conventional ML algorithms as well as RNNs (Khorram and
Jehbez, 2023; Rajesh et al., 2023) and CNNs architectures (Herbert et al., 2021).
However, the biased performance of overfitting, information saturation, and under–fitting issues of ML–
based prediction models has become the challenging issues. It is reviewed that ML models cannot improve the
prediction accuracy without including preprocessing techniques for feature engineering (Apaydin and Sibtain, 2021).
Data preprocessing is conducted at the first step of ML modeling to ensure the data quality for model training. Data
cleaning, transformation, and decomposition are adopted for prediction improvement. In addition, selecting suitable
ML algorithms for specific purpose of hydrological prediction is a challenging task.
This study employed two powerful ML algorithms; XGBoost and LSTM for univariate and multivariate
reservoir inflow prediction of two large storage dams in the Chao Phraya River Basin (CPYRB); (1) Bhumibol (BB)
and Sirikit (SK) dams as shown in Fig. 1. Predicting precise and accurate reservoir inflow for these two reservoirs is
important to ensure reliable water supply sources and establish proper water allocation plan for the downstream water
use in the central region. Furthermore, during the storm seasons from May. to Dec. when the flood risks are likely
elevated, future inflow data is informative for the Office of National Water Resources (ONWR), key decision maker
to implement effective flood mitigation strategies. Input features selection and configuration design of two different
types of daily and monthly reservoir inflow prediction models; (1) univariate prediction with XGBoost and LSTM
algorithms and (2) multivariate prediction with LSTM algorithm, were definitely highlighted. In the last step,
predictability of predicting low and high reservoir inflow values of these models were accordingly explored to assess
their statistical performances.

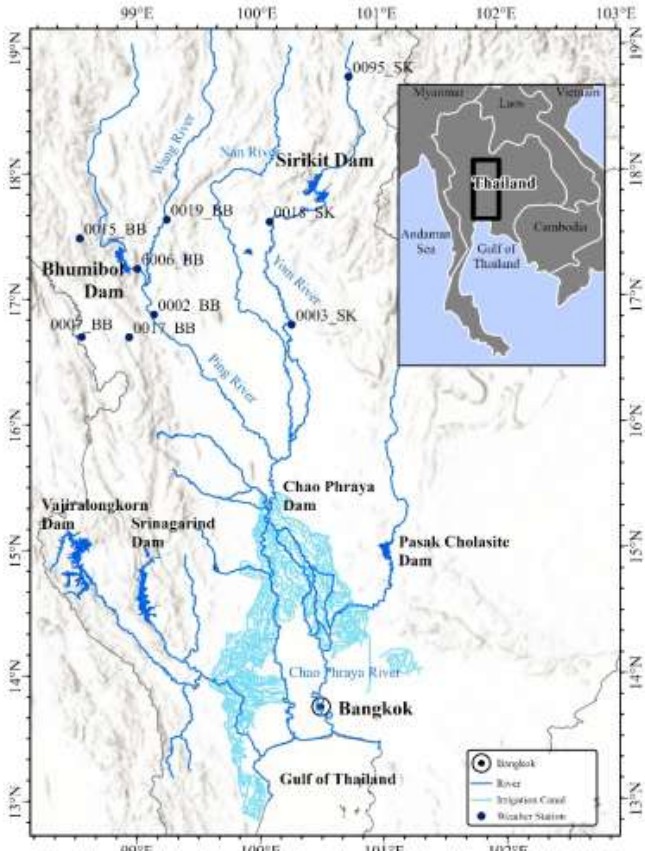

**Fig. 1** Study area in the Chao Phraya River Basin and weather stations






## 2. Material and methods

### *2.1. Input features for reservoir inflow prediction models*

The development of ML prediction models relied on the historical observation data from 2000 to 2020 including reservoir inflows of BB and SK reservoirs, and climate data gathered from the nearest weather stations as summarized in Table 1. As input feature selection is definitely critical to the success of ML– and DL–based prediction models, consequently, the statistical correlation analysis was performed to assess the strength and direction of relationships between climate data from adjacent weather stations and reservoir inflow. By doing this, the daily observed climate data (air humidity, air pressure, temperature, and rainfall) was collected from six Thai Meteorological Department (TMD) stations (0002, 0006, 0007, 0015, 0017, and 0019) located in Tak, Sukhothai, and Lampang provinces near BB dam. Additionally, climate data from the Climate Data Services (CDS) publicly provided by NASA was gathered for locations with geographic coordinates matching those of the TMD stations. Likewise, the daily observed climate data near SK dam was collected from three TMD and NASA stations (0003, 0018, and 0095) located in Phitsanulok, Uttaradit, and Nan provinces, respectively.

For the univariate prediction, single reservoir inflows of BB and SK dams were aimed to individually predict for both daily and monthly models. A number of daily and monthly prediction scenarios with different model configuration using XGBoost and LSTM algorithms were accordingly developed. For the multivariate prediction, reservoir inflows of two main reservoirs, BB–SK dams were expected to achieve simultaneously from single prediction model for both daily and monthly prediction. Consequently, LSTM was selected for multiple output prediction as it can learn and capture the complicated relationship of relevant multiple input variables.

**Table 1** Data used for reservoir inflow prediction modelling

| Data | Description | Data Type | Unit | Data Length |
|---|---|---|---|---|
| Reservoir Inflow_BB | Reservoir Inflow of BB Dam | Daily, Monthly | MCM | 1/1/2000–31/12/2020 |
| Reservoir Inflow_SK | Reservoir Inflow of PS Dam | Daily, Monthly | MCM | 1/1/2000–31/12/2020 |
| Weather | Avg. Humidity at 9 Stations | Daily, Monthly | % | 1/1/2000–31/12/2020 |
| | Avg. Air Pressure at 9 Stations | Daily, Monthly | hPa | 1/1/2000–31/12/2020 |
| | Avg. Temperature at 9 Stations | Daily, Monthly | °C | 1/1/2000–31/12/2020 |
| | Rainfall at 9 Stations at 9 Stations | Daily, Monthly | mm | 1/1/2000–31/12/2020 |
| | Station ID | Station Name | Location: Lat.–Long. | |
| | 0002_BB | Tak | N 16°52'48" – E 99°08'24" | |
| | 0006_BB | Bhumibol Dam | N 17°14'37" – E 99°00'08" | |
| | 0007_BB | Mae Sot | N 16°41'60" – E 98°32'31" | |
| | 0015_BB | Si Samrong | N 17°29'11" – E 99°31'36" | |
| | 0017_BB | Doi Muser | N 16°41'60" – E 98°56'07" | |
| | 0019_BB | Tone | N 17°38'12" – E 99°14'44" | |
| | 0003_SK | Phitsanulok | N 16°47'47" – E 100°16'33" | |
| | 0018_SK | Uttaradit | N 17°37'00" – E 100°05'60" | |
| | 0095_SK | Nan | N 18°46'01" – E 100°45'47" | |

**Note:** meter above mean sea level– m. msl.|Million Cubic Meter–MCM|Cubic Meter per Second–CMS

Selecting input features for each univariate and multivariate prediction model was based on the physical river–reservoir system and statistical cross correlation between past reservoir inflow and relevant climate data as summarized in Table 2. This ensured that the prediction model can effectively capture the strong relationship between the inputs and predicted outputs.





**Table 2** Correlation coefficient between climate data and reservoir inflows of the Bhumibol and Sirikit dams

| Weather Station | Climate Data | TMD Data Source | NASA Data Source | Weather Station | Climate Data | TMD Data Source | NASA Data Source |
|---|---|---|---|---|---|---|---|
| 0002_BB | Avg. humidity (%) | **0.402** | **0.521** | 0019_BB | Avg. humidity (%) | **0.460** | **0.505** |
| | Avg. air pressure (hPa) | –0.117 | –0.148 | | Avg. air pressure (hPa) | –0.090 | –0.146 |
| | Avg. temperature (°C) | –0.115 | –0.190 | | Avg. temperature (°C) | –0.091 | –0.133 |
| | Rainfall (mm/day) | **0.284** | **0.365** | | Rainfall (mm/day) | **0.191** | **0.355** |
| 0006_BB | Avg. humidity (%) | **0.402** | **0.518** | 0003_SK | Avg. humidity (%) | **0.428** | **0.536** |
| | Avg. air pressure (hPa) | –0.007 | –0.146 | | Avg. air pressure (hPa) | –0.328 | –0.322 |
| | Avg. temperature (°C) | –0.103 | –0.173 | | Avg. temperature (°C) | –0.020 | –0.144 |
| | Rainfall (mm/day) | **0.289** | **0.369** | | Rainfall (mm/day) | **0.038** | **0.376** |
| 0007_BB | Avg. humidity (%) | **0.401** | **0.491** | 0018_SK | Avg. humidity (%) | **0.499** | **0.503** |
| | Avg. air pressure (hPa) | –0.164 | –0.179 | | Avg. air pressure (hPa) | –0.023 | –0.339 |
| | Avg. temperature (°C) | –0.096 | –0.076 | | Avg. temperature (°C) | –0.019 | –0.017 |
| | Rainfall (mm/day) | **0.197** | **0.360** | | Rainfall (mm/day) | **0.167** | **0.406** |
| 0015_BB | Avg. humidity (%) | **0.319** | **0.521** | 0095_SK | Avg. humidity (%) | **0.535** | **0.469** |
| | Avg. air pressure (hPa) | –0.021 | –0.139 | | Avg. air pressure (hPa) | –0.379 | –0.358 |
| | Avg. temperature (°C) | –0.027 | –0.178 | | Avg. temperature (°C) | 0.092 | 0.101 |
| | Rainfall (mm/day) | **0.162** | **0.363** | | Rainfall (mm/day) | **0.002** | **0.392** |
| 0017_BB | Avg. humidity (%) | **0.212** | **0.491** | **Note**: | Thai Meteorological Department–TMD| | | |
| | Avg. air pressure (hPa) | 0.003 | –0.479 | | National Aeronautics and Space Administration–NASA | | |
| | Avg. temperature (°C) | 0.006 | –0.076 | | | | |
| | Rainfall (mm/day) | **0.034** | **0.360** | | | | |

The correlation analysis revealed strong correlations between observed reservoir inflows and both humidity
and rainfall at both BB and SK dams. The correlation coefficient between reservoir inflow of BB dam and humidity
data at Station 0006 reaches up to 0.402 and 0.518 for TMD and NASA data sources, respectively. Rainfall data from
Station 0006 also exhibited a strong correlation with reservoir inflow of BB dam, with correlation coefficients of
0.2886 and 0.3693 for TMD and NASA data sources, respectively. The substantial correlation between reservoir
inflow of SK dam and climate data from Station 0018 particularly from NASA data source, was apparently found
with the correlation coefficient of 0.503 and 0.406 for humidity data and precipitation data, respectively. Based on
these analysis, humidity and rainfall data were accordingly selected to specify input structures of ML– and DL–
based prediction models. Autocorrelation of past reservoir inflow was also analyzed to identify optimal lag times and
number of moving average parameters for input feature selection. This analysis revealed the importance of closer lag
time (t to t–7) which exhibited high correlation coefficients exceeding 0.67 with the recent reservoir inflow data for
both BB and SK dams. Accordingly, information of past reservoir inflows with closer lag time t to t–7 was
incorporated into the structure of the prediction models to predict reservoir inflow at lead time t+1.
Following this analysis, four daily and monthly univariate prediction scenarios with various model
configurations varying input features using XGBoost and LSTM algorithms for each of BB and SK dams; S1–S4,
were designed. For the multivariate prediction model, two daily and monthly prediction scenarios; S5–S6, using
LSTM algorithm were established. Major inputs of these prediction models are past reservoir inflow at time t, moving
average of past reservoir inflow at time t–3 and t–7, rainfall at time t, and humidity at time t as summarized in Table
3. It is illustrated that univariate models structured the specific individual inputs for single reservoir inflow prediction
at lead time t+1. For example, the input features of daily reservoir inflow prediction for BB dam are BB past inflow
at time t, moving average of BB past inflow at time t–3 and t–7, rainfall and humidity at time t collected from the
nearest weather stations to BB dam. In contrast, the multivariate prediction models incorporate inputs of both BB and
SK dams to predict two reservoir inflows at lead time t+1.
**Table 3** Input features for univariate and multivariate reservoir inflow prediction models

| Prediction Model | Reservoir Inflow ID | Prediction Scenario | Model Type and Prediction Lead Time | Model No. | Input Features | | | | | |
|---|---|---|---|---|---|---|---|---|---|---|
| | | | | | Past Inflow_BB | Past Inflow_SK | Avg. Past Inflow | Avg. Past Inflow | Rainfall | Humidity |
| – | – | – | t+1 | | t | t | t–3 | t–7 | t | t |
| Univariate Prediction | BB | S1: XGBoost | Daily | dBB–01, dBB–02, dBB–03 | ✓ | – | ✓ | ✓ | ✓ | ✓ |
| | | S2: LSTM | Daily | dBB–01, dBB–02, dBB–03, dBB–04, dBB–05, dBB–06 | ✓ | – | ✓ | ✓ | ✓ | – |
| | SK | S1: XGBoost | Daily | dSK–01, dSK–02, dSK–03 | – | ✓ | ✓ | ✓ | ✓ | ✓ |



| Prediction Model | Reservoir Inflow ID | Prediction Scenario | Model Type and Prediction Lead Time | Model No. | Input Features | | | | | |
|---|---|---|---|---|---|---|---|---|---|---|
| | | | | | Past Inflow_BB | Past Inflow_SK | Avg. Past Inflow | Avg. Past Inflow | Rainfall | Humidity |
| – | – | – | t+1 | | t | t | t–3 | t–7 | t | t |
| | | S2: LSTM | Daily | dSK–01, dSK–02, dSK–03, dSK–04, dSK–05, dSK–06 | – | ✓ | ✓ | ✓ | ✓ | – |
| | BB | S3: XGBoost | Monthly | mBB–01, mBB–02, mBB–03 | ✓ | – | ✓ | ✓ | ✓ | ✓ |
| | | S4: LSTM | Monthly | dBB–01, dBB–02, dBB–03, dBB–04, dBB–05, dBB–06 | ✓ | – | ✓ | ✓ | ✓ | – |
| | SK | S3: XGBoost | Monthly | mSK–01, mSK–02, mSK–03 | – | ✓ | ✓ | ✓ | ✓ | ✓ |
| | | S4: LSTM | Monthly | mSK–01, mSK–02, mSK–03, mSK–04, mSK–05, mSK–06 | – | ✓ | ✓ | ✓ | ✓ | – |
| Multivariate Prediction | BB&SK | S5: LSTM | Daily | dBBSK–01 | ✓ | ✓ | ✓ | – | ✓ | ✓ |
| | BB&SK | S6: LSTM | Monthly | mBBSK–01 | ✓ | ✓ | ✓ | – | ✓ | ✓ |

**Note**: Acronyms–Bhumibol dam–BB|Sirikit dam–SK|Daily Univariate prediction model of BB dam–dBB|Daily univariate prediction model of SK Dam–dSK|Daily multivariate prediction model of BB&SK dams–dBBSK|Monthly multivariate prediction model of BB&SK dams–mBBSK

### 2.2. Prediction algorithms selected
#### 2.2.1. Extreme gradient boosting (XGBoost)

Extreme Gradient Boosting (XGBoost), a powerful machine learning algorithm initiated by Tianqi Chen in 2014 (Chen and Guestrin, 2016), was used to develop the daily and monthly univariate prediction models for reservoir inflow in CPYRB. XGBoost is a decision–tree–based ensemble machine learning as illustrated its structure in Fig. 2. Its efficiency, scalability, and flexibility have been widely demonstrated and proven in hydrological prediction applications (Rajesh et al., 2022). In general, the supervised XGBoost learning primarily involves minimizing objective function which consists of two main components; (1) loss function and (2) regularization term as expressed in Eq. (1). This loss function measures the discrepancy between the predicted and observed values in the model training process as given mean squared error in Eq. (2). The regularization term in Eq. (3) is crucial in preventing model overfitting and complexity for improved prediction performance.

$$Obj(\theta) = L(\theta) + \Omega(\theta) \tag{1}$$

$$L(\theta) = \frac{1}{2}\sum_{i=1}^{n}(y_i - p_i)^2 \tag{2}$$

$$\Omega(\theta) = \gamma T + \frac{1}{2}\lambda \sum_{i=1}^{T} O_{value}^2 \tag{3}$$

where, $L(\theta)$ is the training loss function term. For robust regression tasks, the common loss functions are Mean Squared Error (MSE), Mean Absolute Error (MAE), and Huber loss which is the combination of MSE and MAE. $\Omega(\theta)$ is regularization term. $\theta$ denotes the optimal parameter values that best fits the training inflow data ($y_i$) to the predicted inflow output ($p_i$). $\gamma$ is a hyperparameter controlling the strength of the regularization term which influences the decision to make a further partition on a leaf node of a tree–based model. $T$ is a number of leaf nodes in the tree and $\lambda$ is a hyperparameter used to scale the regularization term. A larger number of leaf nodes signifies the model complexity potentially leading to overfitting. A larger $\lambda$ indicates the increased penalty to model encouraging the reduction of model complexity. $O_{value}$ is a measure of the impurity or heterogeneity of the data points within the leaf node. For tree building process, a prediction for one given data is made by traversing the tree from the root node to a leaf node. The tree is built from a root node and recursively split into left and right child nodes. This process continues until a specific stopping criterion is met as graphically shown in Fig.2. Similarity score (*Sim*) is used to assess the homogeneity of data within a node to guide for leaf node splitting. The larger value of similarity score signifies the similar data within a leaf node that further splitting might not be necessary. Similarity score is computed to indicate a score of each node by using Eq. (4).



$$Sim = \frac{\sum_{i=1}^{n} (y_i - p_i)^2}{n + \lambda} \qquad (4)$$

Gain value is termed to measure the accuracy improvement resulted from a specific splitting. It helps assess
the optimality of potential splits in a tree structure as expressed in Eq. (5). A higher positive gain value indicates a
better split in improving the model predictive performance. When the gain values are negative, the tree branch is
removed as shown in Fig. 3.

$$Gain\ value = (Simleft + Simright) - Simroot \qquad (5)$$

where, *Simleft*, *Simright*, and *Simroot* denote the similarity score of the left leaf node, right leaf node, and
root node of the branch, respectively. The tree structures are iteratively built for T iterations until the desired number
of models is reached. In each iteration, the output value ($O_{value}$) for all leaf nodes is computed using Eq. (6).

$$O_{value} = \frac{\sum_{i=1}^{n} (y_i - p_i)}{n + \lambda} \qquad (6)$$

In addition, the precision and speed of convergence of the prediction model is governed by learning rate ($\varepsilon$).
Learning rate determines level of model improvement to handle the prediction error made by previous iterations. A
larger values of learning rate can accelerate the training process leading to faster convergence. However, overfitting
can simply find if not properly fine–tuned. In contrast, the smaller values of learning rate can help reduce overfitting
but speed of convergence is definitely lower. In the final step, XGBoost can make updated prediction ($p_i^t$) by combining
the initial prediction ($p_i^0$) with the gradient of the loss function and the regularization term multiplied with learning
rate as expressed in Eq. (7).

$$p_i^t = p_i^0 + \varepsilon [\sum_{i=1}^{n} L (y_i, p_i^0 + O_{value}) + \frac{1}{2}\lambda O_{value}^2] \qquad (7)$$


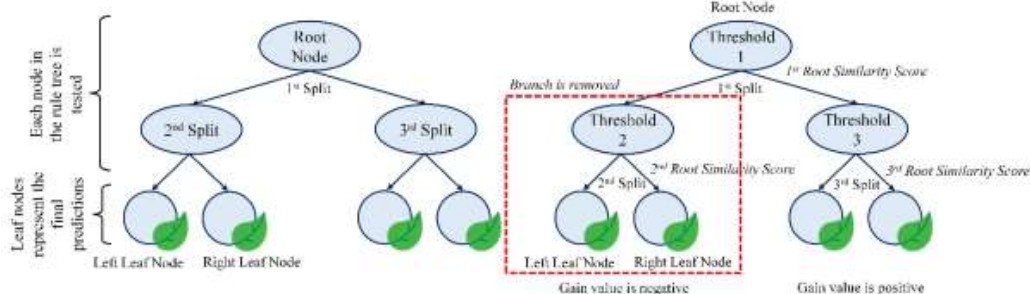

**Fig. 2** XGBoost tree–based structure

***2.2.2. Long Short–Term Memory (LSTM)***
Long Short–Term Memory (LSTM) is a well–suited type of deep learning algorithm designed to process
sequential data. It was initially introduced by Hochreiter and Schmidhuber in 1997 (Hochreiter and Schmidhuber,
1997). LSTM is an evolution of Recurrent Neural Networks (RNN) which is a type of Artificial Neural Networks
(ANNs) having memory to process sequential inputs. However, the complications to learn and capture long–term
dependencies due to vanishing and exploding gradient problems become the significant drawback of RNN. To
overcome this, LSTM is specifically developed to learn long–term dependencies and retain previous information over
extended periods. The LSTM model is commonly structured as a chain of units as illustrated in Fig. 3. Each LSTM
unit is composed of a cell state and three gates namely, (1) input gate, (2) forget gate, and output gate. Cell state is
functioned as core component of LSTM to store and carry information through time steps. Input gate regulates new
information from current input to be stored in a cell state. Forget gate decides to discard or keep information from
previous cell state. Output gate determines which part of cell state should be the current prediction output (Khorram
and Jehbez, 2023; Rajesh et al., 2023).



In the initial step, forget gate examines the current input at time t ($x_t$), previous hidden state ($h_{t-1}$) which is the output at previous time t–1, and long–term memory from previous cell state at time t–1 ($C_{t-1}$). It calculates a value in a range of 0 and 1 for each element in the previous cell state as expressed in Eq. (8). Subsequently, the input gate calculates two values; input gate at time t ($i_t$) and cell state input at time t ($Č_t$) to regulate the new information into the current cell state as defined in Eq. (9) and Eq. (10). Then, the new cell state at time t ($C_t$) in Eq. (11) is updated by combining the new information of previous cell state at time t–1 ($C_{t-1}$), forget gate at time t ($f_t$), and input gate at time t ($i_t$, $Č_t$). Eq. (12) is employed to compute the current prediction output at time t ($o_t$). The hidden state ($h_t$) indicating LSTM output at time t, is finally calculated by multiplying the output gate at time t ($o_t$) with the tanh of the new cell state at time t ($C_t$) as shown in Eq. (12) and Eq. (13). By doing this through iteration process, LSTM can handle the sequential data and make accurate and precise prediction.

$$f_t = \sigma(W_f * [h_{t-1}, x_t] + b_f) \tag{8}$$

$$i_t = \sigma(W_i * [h_{t-1}, x_t] + b_i) \tag{9}$$

$$Č_t = tanh(W_C * [h_{t-1}, x_t] + b_C) \tag{10}$$

$$C_t = f_t * C_{t-1} + i_t * Č_t \tag{11}$$

$$o_t = \sigma(W_o * [h_{t-1}, x_t] + b_f) \tag{12}$$

$$h_t = o_t * tanh(C_t) \tag{13}$$

Where $W_{f/i/C}$ is corresponding weight matrix, $b_{f/i/C}$ is corresponding bias and $\sigma$ is the sigmoid activation function.

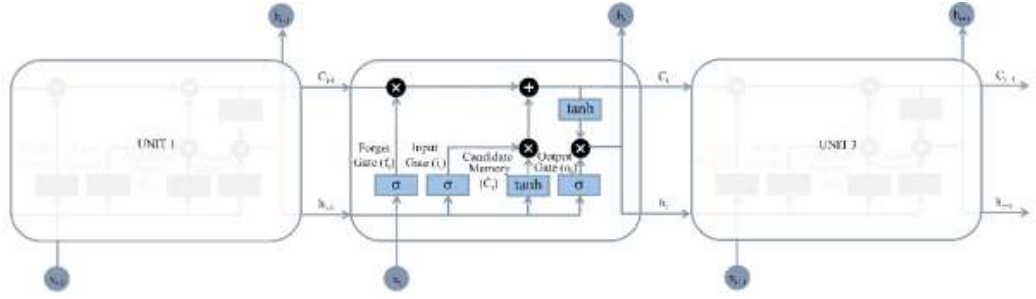

**Fig. 3** LSTM chain–like structure and its unit

### *2.3. Model configuration design and prediction modelling*

Model configuration design of daily and monthly univariate and multivariate prediction models are presented in Fig. 4 and Table 4–Table 5. The input features as aforementioned in Section 2.1., designated training–testing dataset ratios (60:40, 70:30, 80:20), delayed time of moving average inflow (t–3 and t–7), and learning rates (0.001, 0.01, 0.1) were varied to optimize the model configuration and prediction performance. Implementation of training model for daily and monthly prediction by XGBoost was controlled by the hyperparameter tunning process such as gamma, maximum depth of a tree, number of iterations (nrounds), number of threads (nthreads), learning rate, number of fold (nfolds) and early stopping rounds (early_stopping_rounds) parameters as presented in Table 4. In this study, maximum depth of a tree was set to 6 in increase deeper tree model complexity. Maximum number of iterations was specified to 10,000 for model training process. Number of threads determining for parallel computation to speed up the training process and number of folds specifying for cross–validation was 10 and 2, respectively. The early stopping rounds were generally used to stop training procedures when the loss on training dataset starts increasing. In this study, the early stopping round was set every 500 iterations if the performance on RMSE was not substantially improved.



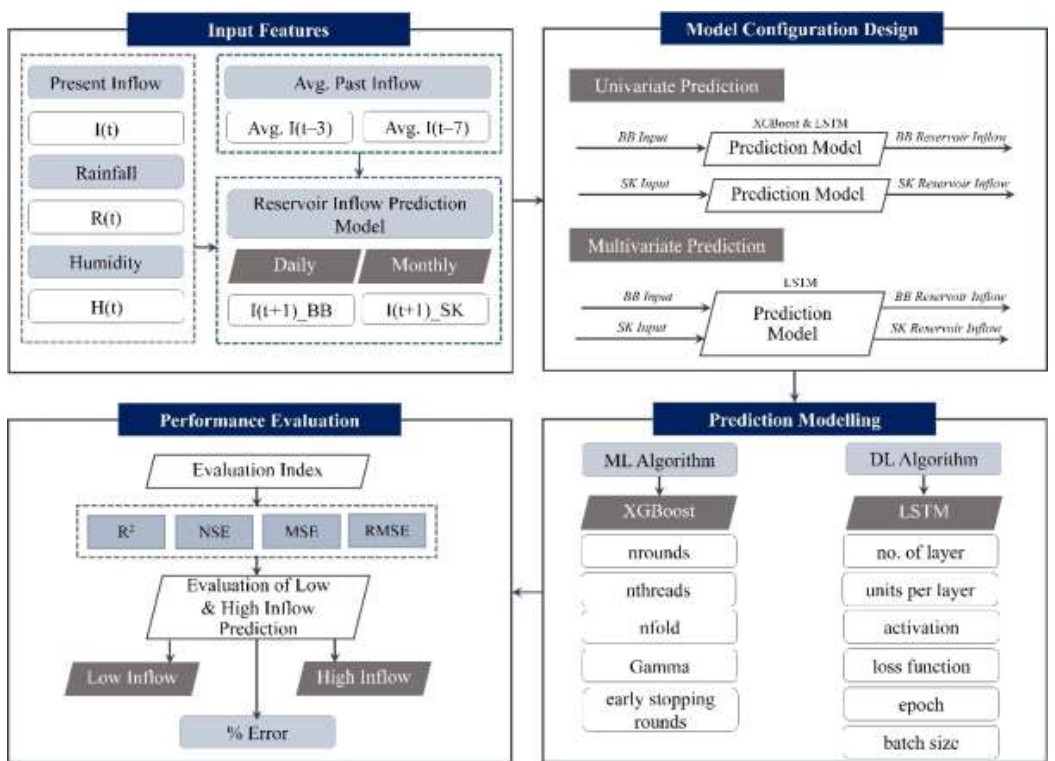

**Fig. 4** Workflow diagram of this study


**Table 4** Model configuration design of daily and monthly univariate prediction models using XGBoost

| Model No. | Input Feature | Training:Testing Ratio | Learning Rate | XGBoost Parameter | | | | | | | |
|-----------|---------------|------------------------|---------------|-------------------|-------|--------------|------------|---------|----------|--------|----------------------------|
| | | | | Objective | Gamma | Max. Depth | Evaluation | nrounds | nthreads | nfolds | Early stopping rounds |
| colspan: Daily and Monthly Prediction of BB Reservoir Inflow |||||||||||
| dBB–01, mBB–01 | Past Inflow, Avg. Past Inflow | 60:40, 70:30, 80:20 | 0.001, 0.01, 0.1 | regression: linear | 0 | 6 | RMSE | 10,000 | 10 | 2 | 500 |
| dBB–02, mBB–02 | Past Inflow, Avg. Past Inflow, Rainfall | 60:40, 70:30, 80:20 | 0.001, 0.01, 0.1 | regression: linear | 0 | 6 | RMSE | 10,000 | 10 | 2 | 500 |
| dBB–03, mBB–03 | Past Inflow, Avg. Past Inflow, Rainfall, Humidity | 60:40, 70:30, 80:20 | 0.001, 0.01, 0.1 | regression: linear | 0 | 6 | RMSE | 10,000 | 10 | 2 | 500 |
| colspan: Daily and Monthly Prediction of SK Reservoir Inflow |||||||||||
| dSK–01, mSK–01 | Past Inflow, Avg. Past Inflow | 60:40, 70:30, 80:20 | 0.001, 0.01, 0.1 | regression: linear | 0 | 6 | RMSE | 10,000 | 10 | 2 | 500 |
| dSK–02, mSK–02 | Past Inflow, Avg. Past Inflow, Rainfall | 60:40, 70:30, 80:20 | 0.001, 0.01, 0.1 | regression: linear | 0 | 6 | RMSE | 10,000 | 10 | 2 | 500 |
| dSK–03, mSK–03 | Past Inflow, Avg. Past Inflow, Rainfall, Humidity | 60:40, 70:30, 80:20 | 0.001, 0.01, 0.1 | regression: linear | 0 | 6 | RMSE | 10,000 | 10 | 2 | 500 |






Similarly, model configuration using LSTM for both univariate and multivariate prediction models was
designed by varying input features and tuning key hyperparameters such as number of layers, number of units per
layer, activation function, optimizer, epoch, and batch size, etc. as summarized in Table 5. In this study, number of
layers was set to 2 and units per layer referring number of neurons in a specific layer of LSTM network was set to 64
or 32. Rectified Linear Unit (ReLU), a famous activation function used in deep neural networks was specified in all
univariate prediction experiments while Sigmoid, a traditional activation function was used for multivariate prediction.
Adam optimizer which is an optimization algorithm, was employed to update weights during training process. MSE
was used as loss function for both univariate and multivariate prediction models. Additionally, number of epochs
involving the passing numbers through the entire training dataset was varied between 50, 100, and 500. Similarly,
batch size incorporating number of samples processed before updating the model's weights was set to 16, 32, and 64.
**Table 5** Model configuration design of daily and monthly univariate and multivariate prediction models using LSTM

| Model No. | Input Feature | Standardization Technique | LSTM Parameter in Keras | | | | | | | |
|---|---|---|---|---|---|---|---|---|---|---|
| | | | Steps | Number of Layers | Units per Layer | Activation | Loss Function | Optimizer | Epoch | Batch Size |
| Daily Univariate Prediction Models of BB Dam | | | | | | | | | | |
| dBB–01 | Past Inflow, Avg. Past Inflow | Standard | 3 | 2 | 64/32 | ReLU | MSE | Adam | 50 | 16 |
| dBB–02 | Past Inflow, Avg. Past Inflow | Standard | 7 | 2 | 64/32 | ReLU | MSE | Adam | 50 | 16 |
| dBB–03 | Past Inflow, Avg. Past Inflow | Standard | 14 | 2 | 64/32 | ReLU | MSE | Adam | 50 | 16 |
| dBB–04 | Past Inflow, Avg. Past Inflow, Rainfall | Standard | 3 | 2 | 64/32 | ReLU | MSE | Adam | 50 | 16 |
| dBB–05 | Past Inflow, Avg. Past Inflow, Rainfall | Standard | 7 | 2 | 64/32 | ReLU | MSE | Adam | 50 | 16 |
| dBB–06 | Past Inflow, Avg. Past Inflow, Rainfall | Standard | 14 | 2 | 64/32 | ReLU | MSE | Adam | 50 | 16 |
| Daily Univariate Prediction Models of SK Dam | | | | | | | | | | |
| dSK–01 | Past Inflow, Avg. Past Inflow | Standard | 3 | 2 | 64/32 | ReLU | MSE | Adam | 50 | 16 |
| dSK–02 | Past Inflow, Avg. Past Inflow | Standard | 7 | 2 | 64/32 | ReLU | MSE | Adam | 50 | 16 |
| dSK–03 | Past Inflow, Avg. Past Inflow | Standard | 14 | 2 | 64/32 | ReLU | MSE | Adam | 50 | 16 |
| dSK–04 | Past Inflow, Avg. Past Inflow, Rainfall | Standard | 3 | 2 | 64/32 | ReLU | MSE | Adam | 50 | 16 |
| dSK–05 | Past Inflow, Avg. Past Inflow, Rainfall | Standard | 7 | 2 | 64/32 | ReLU | MSE | Adam | 50 | 16 |
| dSK–06 | Past Inflow, Avg. Past Inflow, Rainfall | Standard | 14 | 2 | 64/32 | ReLU | MSE | Adam | 50 | 16 |
| Monthly Univariate Prediction Models of BB Dam | | | | | | | | | | |
| mBB–01 | Past Inflow, Avg. Past Inflow | MinMax | 3 | 2 | 64/32 | ReLU | MSE | Adam | 100 | 16 |
| mBB–02 | Past Inflow, Avg. Past Inflow | MinMax | 3 | 2 | 64/32 | ReLU | MSE | Adam | 100 | 32 |
| mBB–03 | Past Inflow, Avg. Past Inflow | MinMax | 3 | 2 | 64/32 | ReLU | MSE | Adam | 100 | 64 |
| mBB–04 | Past Inflow, Avg. Past Inflow, Rainfall | MinMax | 3 | 2 | 64/32 | ReLU | MSE | Adam | 100 | 16 |
| mBB–05 | Past Inflow, Avg. Past Inflow, Rainfall | MinMax | 3 | 2 | 64/32 | ReLU | MSE | Adam | 100 | 32 |
| mBB–06 | Past Inflow, Avg. Past Inflow, Rainfall | MinMax | 3 | 2 | 64/32 | ReLU | MSE | Adam | 100 | 64 |
| Monthly Univariate Prediction Models of SK Dam | | | | | | | | | | |
| mSK–01 | Past Inflow, | MinMax | 3 | 2 | 64/32 | ReLU | MSE | Adam | 100 | 16 |




| Model No. | Input Feature | Standardization Technique | LSTM Parameter in Keras | | | | | | | | |
|---|---|---|---|---|---|---|---|---|---|---|---|
| | | | Steps | Number of Layers | Units per Layer | Activation | Loss Function | Optimizer | Epoch | Batch Size |
| | Avg. Past Inflow | | | | | | | | | |
| mSK–02 | Past Inflow, Avg. Past Inflow | MinMax | 3 | 2 | 64/32 | ReLU | MSE | Adam | 100 | 32 |
| mSK–03 | Past Inflow, Avg. Past Inflow | MinMax | 3 | 2 | 64/32 | ReLU | MSE | Adam | 100 | 64 |
| mSK–04 | Past Inflow, Avg. Past Inflow, Rainfall | MinMax | 3 | 2 | 64/32 | ReLU | MSE | Adam | 100 | 16 |
| mSK–05 | Past Inflow, Avg. Past Inflow, Rainfall | MinMax | 3 | 2 | 64/32 | ReLU | MSE | Adam | 100 | 32 |
| mSK–06 | Past Inflow, Avg. Past Inflow, Rainfall | MinMax | 3 | 2 | 64/32 | ReLU | MSE | Adam | 100 | 64 |
| Daily and Monthly Multivariate Prediction Models of BB–SK Dam | | | | | | | | | | |
| dBBSK–01, mBBSK–01, | BB Past Inflow, SK Past Inflow, BB Avg. Past Inflow SK Avg. Past Inflow, BB Rainfall, SK Rainfall, BB Humidity, SK Humidity | MinMax | 3 | 2 | 64 | Sigmoid | MSE | Adam | 500 | 64 |

### 2.4. Evaluation of predictive performance

The statistical metrics; Coefficient of Determination ($R^2$), Nash–Sutcliffe Efficiency (NSE), Mean Squared Error (MSE), Root Mean Squared Error (RMSE), were used to evaluate the perfect match between the predicted and observation values of reservoir inflows. $R^2$ is statistical measures describing the degree of linear correlation between two independent variables which ranges from 0 to 1 (Al–Aqeeli et al., 2015). A higher $R^2$ value closer to 1 indicates better fit of the prediction model to the observation values making stronger predictive power. NSE is the normalized statistical measure that compares the relative magnitude of the model prediction errors to observed data variance ranging from –∞ to 1 (Brownlee, 2018). The prediction accuracy can be classified into three main classes subject to NSE. When NSE is greater than or equal to 0.90, the prediction accuracy is classified as "Class A–Excellent", NSE ranges between 0.70 and 0.90, it is considered as "Class B–Good), and NSE lies between 0.50 and 0.70, it is classified as "Class C–Moderate" (China National Standardization Management Committee, 2008; Zhang et al., 2021). MSE, and RMSE metrics quantify the absolute and squared differences between the predicted and actual values, respectively. A lower value of MSE, and RMSE indicates better model performance. A prediction model is considered as precise and robust prediction when $R^2$ and NSE values are relatively approach to 1, MSE and RMSE values are small.

In the last step, the predictability to predict the low, average, and high daily and monthly reservoir inflows of BB and SK dams was assessed to leverage the application of ML–and DL–based prediction model for real–time reservoir operation and planning during the critical periods. The lowest, average, and highest reservoir inflows of the tested results were compared to the observed reservoir inflows. Finally, percentage error in prediction was computed.

## 3. Results and Discussion
### 3.1. Predicted one–day and one–month ahead of BB and SK reservoir inflows

In this study, $R^2$ and NSE were prioritized over MSE and RMSE to identify the optimal predictive model performances on the training and testing datasets. The quantitative and qualitative comparisons between observed and predicted inflows of the optimal daily and monthly univariate and multivariate prediction models for six scenarios are presented in Table 6 and illustrated in Fig. 5 and Fig. 6. The qualitative results from 2000 to 2020 demonstrated that daily and monthly predicted inflows for both BB and SK dams closely matched the observed inflows for both training and testing datasets when univariate and multivariate prediction models were performed. Even the monthly inflow pattern obtained from univariate and multivariate prediction models are likely similar to observed values, however, monthly prediction exhibited larger deviation from the observed values compared to daily prediction particularly the lowest and highest reservoir inflows.



**Table 6** The optimal predictive performances of daily and monthly reservoir inflow prediction

| Univariate Prediction Model | | | | | Multivariate Prediction Model | | | | |
|---|---|---|---|---|---|---|---|---|---|
| Scenario Design | Model Building | Statistical Metrics | BB | SK | Scenario Design | Model Building | Statistical Metrics | BB | SK |
| S1: Daily model using XGBoost algorithm | Training | $R^2$ | 0.922 | 0.884 | S5: Daily model using LSTM algorithm | Training | $R^2$ | 0.894 | 0.842 |
| | | NSE | 0.909 | 0.871 | | | NSE | 0.890 | 0.841 |
| | | MSE | 62.919 | 70.000 | | | MSE | 75.698 | 97.183 |
| | | RMSE | 7.932 | 8.367 | | | RMSE | 8.700 | 9.858 |
| | Testing | $R^2$ | 0.885 | 0.836 | | Testing | $R^2$ | 0.873 | 0.780 |
| | | NSE | 0.862 | 0.816 | | | NSE | 0.857 | 0.767 |
| | | MSE | 31.990 | 81.308 | | | MSE | 36.039 | 92.455 |
| | | RMSE | 5.656 | 9.017 | | | RMSE | 6.003 | 9.615 |
| S2: Daily model using LSTM algorithm | Training | $R^2$ | 0.925 | 0.878 | S6: Monthly model using LSTM algorithm | Training | $R^2$ | 0.542 | 0.518 |
| | | NSE | 0.924 | 0.878 | | | NSE | 0.538 | 0.512 |
| | | MSE | 46.898 | 61.577 | | | MSE | 183,353 | 192,778 |
| | | RMSE | 6.848 | 7.847 | | | RMSE | 428 | 439 |
| | Testing | $R^2$ | 0.827 | 0.851 | | Testing | $R^2$ | 0.526 | 0.487 |
| | | NSE | 0.818 | 0.851 | | | NSE | 0.397 | 0.459 |
| | | MSE | 60.770 | 67.050 | | | MSE | 103,050 | 130,290 |
| | | RMSE | 7.800 | 8.190 | | | RMSE | 321 | 361 |
| S3: Monthly model using XGBoost algorithm | Training | $R^2$ | 0.452 | 0.490 | | | | | |
| | | NSE | 0.411 | 0.473 | | | | | |
| | | MSE | 217,267 | 196,562 | | | | | |
| | | RMSE | 466 | 443 | | | | | |
| | Testing | $R^2$ | 0.679 | 0.520 | | | | | |
| | | NSE | 0.675 | 0.513 | | | | | |
| | | MSE | 65,836 | 128,363 | | | | | |
| | | RMSE | 257 | 358 | | | | | |
| S4: Monthly model using LSTM algorithm | Training | $R^2$ | 0.519 | 0.678 | | | | | |
| | | NSE | 0.513 | 0.673 | | | | | |
| | | MSE | 186,719 | 104,617 | | | | | |
| | | RMSE | 432 | 323 | | | | | |
| | Testing | $R^2$ | 0.388 | 0.434 | | | | | |
| | | NSE | 0.353 | 0.407 | | | | | |
| | | MSE | 122,597 | 158,222 | | | | | |
| | | RMSE | 350 | 398 | | | | | |



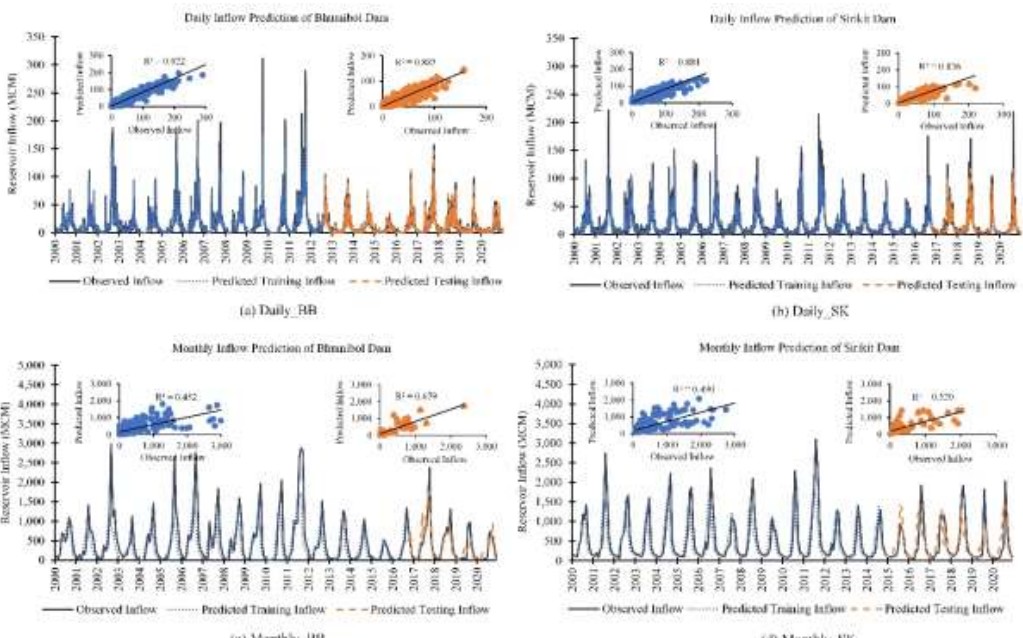

**Fig. 5** Optimal predictive performances of one–day and one–month ahead univariate prediction models of BB and SK reservoir inflows

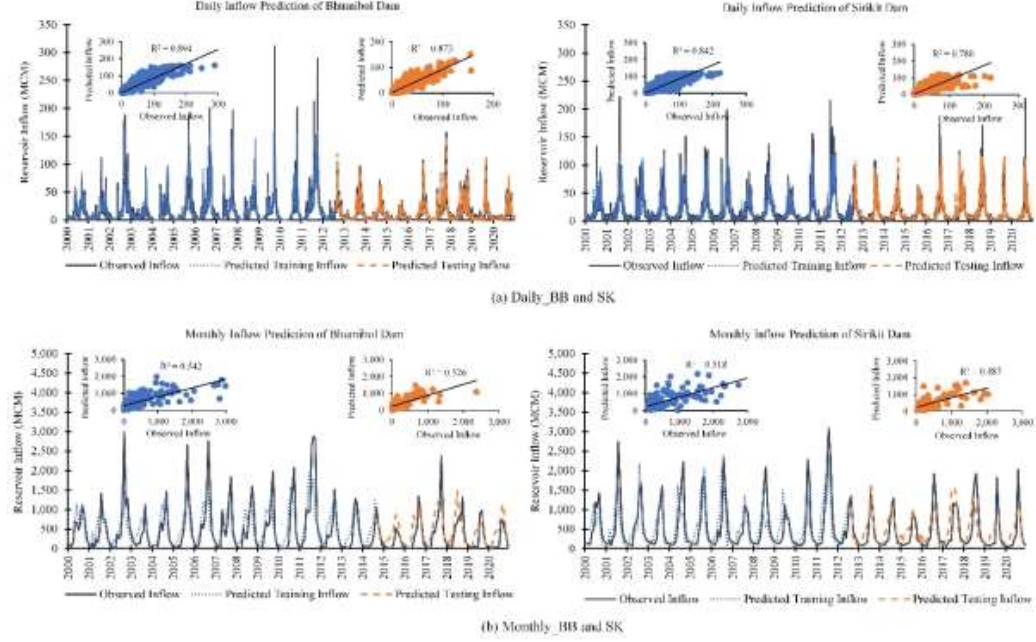

**Fig. 6** Optimal predictive performances of one–day and one–month ahead multivariate prediction models of BB and SK reservoir inflows



Fig. 7 and Fig. 8 quantitatively illustrate the performance metrics of all scenarios of 1–day ahead and 1–month ahead reservoir inflow predictions for both univariate and multivariate predictions. For the daily univariate prediction using XGBoost on the training dataset, Scenario 1 (S1) achieved $R^2$ and NSE values of 0.922 and 0.909 for BB dam and 0.884 and 0.871 for SK dam, respectively. However, $R^2$ and NSE values were slightly lower on the testing dataset with the values of 0.885 and 0.862 for BB dam and 0.836 and 0.816 for SK dam, respectively. MSE and RMSE values for BB dam were likely lower by –49.16% and –28.68% when 20% of dataset was accordingly tested. In contrast, these two values were slightly increased on the testing dataset for SK dam by +16.15% and +7.77%. It is revealed that the daily univariate prediction using LSTM (Scenario 2, S2) on the training dataset mostly demonstrated higher $R^2$ and NSE values of 0.925 and 0.924 for BB dam and 0.878 and 0.878 for SK dam, respectively indicating slightly higher predictive performances than XGBoost. Similar to XGBoost, $R^2$ and NSE values slightly decreased on the testing dataset achieving 0.827 and 818 for BB dam. For SK dam, these two statistical metrics considerably increased to 0.851 and 0.851 for SK dam, respectively. However, MSE and RMSE values performed by LSTM were +29.58% and +13.90% for BB dam and +8.89% and +4.37% for SK dam which were lower than XGBoost, respectively. For the Scenario 5 (S5) when daily multivariate prediction model was executed using LSTM to predict reservoir inflow for both BB and SK dams, $R^2$ and NSE values of training dataset were 0.894 and 0.890 for BB dam and 0.842 and 0.841 for SK dam, respectively. These values were slightly lower than those trained by daily univariate prediction model using both XGBoost and LSTM. On the testing dataset with daily multivariate prediction model, $R^2$ and NSE values for BB dam were slightly higher than those obtained by daily univariate prediction model for both XGBoost and LSTM, reaching 0.873 and 0.857, respectively. However, these values decreased considerably to 0.780 and 0.767, respectively for SK dam. In terms of MSE and RMSE, there was an insubstantial difference between univariate and multivariate prediction models.

The predictive results exhibited that the predictability of daily univariate prediction models is slightly superior than multivariate models, as they were developed independently to extract the specific input features of each dam. However, the daily multivariate prediction model could provide two predictive outputs relatively closer to daily univariate prediction model which is beneficial for real–time operational applications in the river basin. Due to the high complexity of all input features and larger dataset used in the daily multivariate prediction model, however, DL–based prediction model with LSTM algorithm could perform well in learning and capturing the input–output relation and features, resulting in providing predictive performance closer to those achieved by the daily univariate prediction model. Training larger datasets with LSTM for both daily univariate and multivariate prediction models gave the better predictive performances than XGBoost. However, when smaller testing datasets were tested by LSTM, statistical performance was slightly decreased compared to training dataset as obviously presented in Scenario 2 and Scenario 5. It is also revealed that daily reservoir inflow predictions for BB dam demonstrated higher performance compared to SK dam, for both daily univariate and multivariate models. In comparison between XGBoost and LSTM for daily reservoir inflow prediction on tested datasets, individual XGBoost model outperformed LSTM models across both univariate and multivariate predictions for BB dam. For SK dam, the individual LSTM prediction model could predict better results than the individual XGBoost model and the LSTM multivariate prediction model.



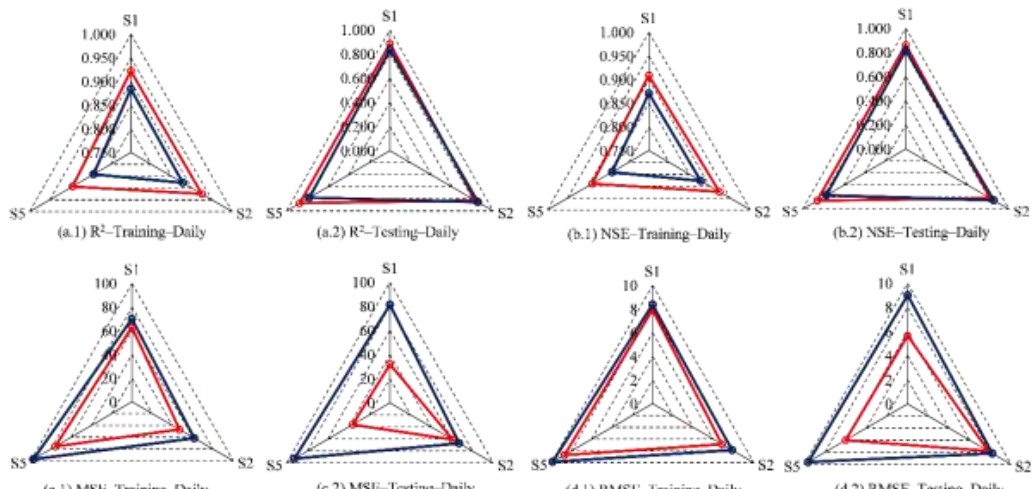

**Fig. 7** Radar chart illustrating the performance metrics of all scenarios of 1–day ahead reservoir inflow prediction (Legend: Red–BB, Blue–SK)


For ML–based monthly univariate prediction using XGBoost, Scenario 3 (S3) achieved $R^2$ and NSE values
of 0.452 and 0.411 for BB dam and 0.490 and 0.473 for SK dam, respectively on the training dataset. In contrast to
daily prediction models with XGBoost, $R^2$ and NSE values were slightly higher on the testing dataset with the values
of 0.679 and 0.675 for BB dam and 0.520 and 0.513 for SK dam, respectively. However, MSE and RMSE values were
significantly decreased by –69.70% and –44.85% for BB dam and –34.70% and –19.19% for SK dam when testing
dataset was accordingly employed. DL–based monthly univariate prediction using LSTM in Scenario 4 (S4) exhibited
higher statistical performances in terms of $R^2$ and NSE values of 0.519 and 0.513 for BB dam and 0.678 and 0.673 for
SK dam, respectively. However, these performance metrics considerably decreased to 0.388 and 0.353 for BB dam
and 0.434 and 0.407 for SK dam when the testing datasets were investigated. The MSE and RMSE values performed
by monthly prediction model with LSTM were significantly decreased by –34.34% and –18.98% for BB dam. In
contrast, it showed the substantial increase in MSE and RMSE values by +53.24% and +23.23% for SK dam when
testing dataset was accordingly employed. Compared to the individual monthly prediction by LSTM on testing
datasets, LSTM–based monthly multivariate prediction in Scenario 6 (S6) could predict better results achieving $R^2$
and NSE values of 0.526 and 0.397 for BB dam and 0.487 and 0.459 for SK dam, respectively. However, it showed
lower $R^2$ and NSE values compared to those obtained by individual monthly prediction using XGBoost.
In comparison between daily and monthly prediction models, the results demonstrated that ML– and DL–
based prediction models achieved higher statistical metrics in predicting daily reservoir inflow compared to monthly
predictions. This is because prediction modeling with ML and DL algorithms can handle and leverage larger datasets
available in daily prediction model, enabling them to learn and capture patterns more effectively. Similarly, training
larger datasets with LSTM for both daily and monthly univariate and multivariate prediction models mostly
outperformed XGBoost. However, slight decrease in statistical metrics is found when dealing with the smaller testing
datasets. To improve the robustness and precision of LSTM–based forecasts for both univariate and multivariate
predictions, it is recommended to increase the testing dataset size during model validation. Importantly, cross–
validation should be conducted to assess model overfitting and ensure its generalizability.





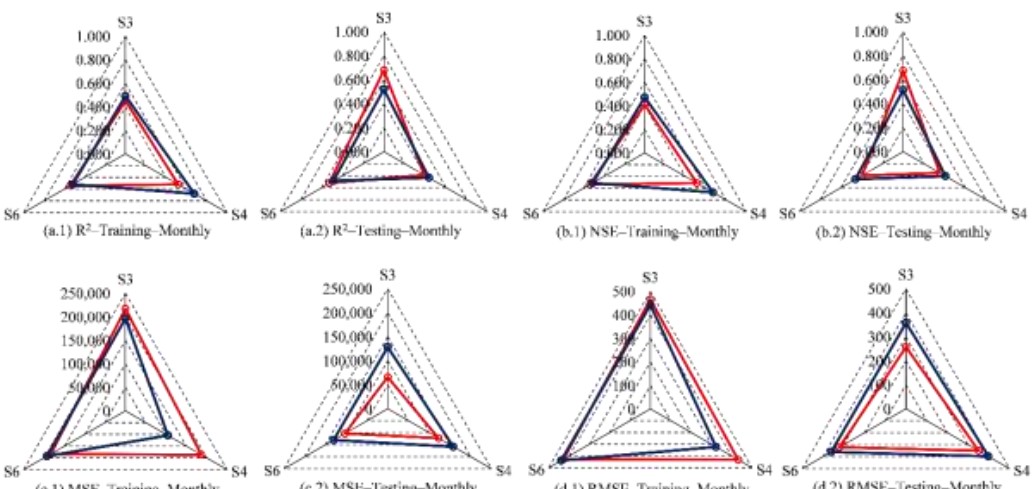

**Fig. 8** Radar chart illustrating the performance metrics of all scenarios of 1–month ahead reservoir inflow prediction (Legend: Red–BB, Blue–SK)

### 3.2. Optimal model configuration for univariate and multivariate reservoir inflow prediction models

Based on the optimization criteria of maximizing $R^2$ and NSE values as optimal prediction model, accordingly the optimal input features, training and testing dataset ratio, and optimal hyperparameter of prediction models were explored. The results of optimal model configuration are summarized in Table 7.

**Table 7** Optimal model configuration for reservoir inflow prediction models

| Input Features & Training:Testing Ratio & Hyperparameter | S1: Daily Univariate Model_XGBoost | | S2: Daily Univariate Model_LSTM | | S3: Monthly Univariate Model_XGBoost | | S4: Monthly Univariate Model_LSTM | | S5: Daily Multivariate Model_LSTM | S6: Monthly multivariate Model_LSTM |
|---|---|---|---|---|---|---|---|---|---|---|
| | BB | SK | BB | SK | BB | SK | BB | SK | BB–SK | BB–SK |
| Past Inflow at Time t | ✓ | ✓ | ✓ | ✓ | ✓ | ✓ | ✓ | ✓ | ✓ | ✓ |
| Avg. Past Inflow at t–3 | ✓ | ✓ | – | – | ✓ | – | ✓ | ✓ | ✓ | ✓ |
| Avg. Past Inflow at t–7 | – | – | ✓ | ✓ | – | ✓ | – | – | – | – |
| Rainfall at Time t | – | – | – | – | ✓ | ✓ | – | – | ✓ | ✓ |
| Humidity at Time t | – | – | – | – | ✓ | – | – | – | ✓ | ✓ |
| Training:Testing Ratio | 80:20 | 80:20 | **70:30** | **70:30** | 80:20 | 70:30 | **60:40** | **60:40** | **70:30** | **60:40** |
| XGBoost: Learning Rate | 0.1 | 0.1 | – | – | 0.001 | 0.001 | – | – | – | – |
| XGBoost: Nrounds | 10,000 | 10,000 | – | – | 10,000 | 10,000 | – | – | – | – |
| XGBoost: Max Depth | 6 | 6 | – | – | 6 | 6 | – | – | – | – |
| LSTM: Learning Rate | – | – | 0.1 | 0.1 | – | – | 0.1 | 0.1 | 0.1 | 0.1 |
| LSTM: No. of Layers | – | – | 2 | 2 | – | – | 2 | 2 | 2 | 2 |
| LSTM: No. of Units | – | – | 64 | 64 | – | – | 64 | 64 | 64 | 64 |

It is emphasized that information on past reservoir inflow including the previous time step and its moving average was observed to be a significant predictor to predict future inflow for all prediction scenarios. For daily and monthly multivariate predictions by LSTM, rainfall and humidity data was incorporated as input features which demonstrated a substantial impact on predictive performances. Since XGBoost can work effectively with smaller datasets compared to LSTM for both univariate and multivariate prediction models, it is crucial to select key predictors that are highly correlated with reservoir inflow for the model development. Conversely, LSTM is specifically designed for larger sequential datasets, a broad range of potential input features can be included in the prediction models. Moreover, altering optimal training and testing ratios by increasing the testing datasets into 70:30 and 60:40 can significantly enhance the predictive performances of optimal LSTM–based prediction models. However, handling the larger datasets require more computational resource to capture relevant data features making LSTM–based prediction models computationally expensive than XGBoost. Furthermore, multivariate prediction models generally consume more computational resources than univariate prediction models.



It is revealed from the model configuration experiment that learning rate of 0.1 and 0.001 significantly
impacts stability and convergence of both XGBoost and LSTM algorithms during model training. The number of
boosting rounds (nrounds) of 10,000 in all scenarios of XGBoost–based prediction models can improve the statistical
performance metrics substantially. However, overfitting risk due to increased number of boosting rounds should be
carefully investigated through model validation. In this study, to control complexity of individual tree and capture
complex data pattern from deeper tress, maximum depth was set to 6 for all scenarios of univariate prediction. For the
LSTM–based prediction models, learning rate of 0.1 of both univariate and multivariate prediction models is a
significant hyperparameter in achieving model stability and convergence. Number of layers was set to 2 across all
prediction scenarios to identify model complexity. Furthermore, the optimal number of LSTM units was observed to
be 64 across all scenarios exhibiting increased predictive performances.
### *3.3. Evaluation of Percentage Error of Low, Average and High Reservoir Inflow Prediction*
The percentage error of low, average and high reservoir inflow prediction, was computed to diagnose the
predictability of ML– and DL–based prediction models as summarized the results in Table 8. It is revealed that the
daily minimum and average reservoir inflows performed by XGBoost univariate model (S1) on the training and testing
datasets were very close to the observed values. On the training datasets, a small discrepancy of daily minimum and
average reservoir inflows was +0.17 MCM and –0.81 MCM (–4.62% error), respectively for BB dam and +3.03 MCM
and –0.62 MCM or –3.58%, respectively for SK dam. Similarly, it showed small discrepancy of daily minimum and
average reservoir inflows on the testing datasets was +0.17 MCM and +0.03 MCM (+0.27% error), respectively for
BB dam and +3.03 MCM and –0.34 MCM or –2.38%, respectively for SK dam. Importantly, this analysis exhibited
a consistent overprediction of minimum reservoir inflows for these two dams on both training and testing datasets. In
contrast, larger percentage error in underprediction of maximum reservoir inflows on both training and testing datasets
were observed ranging –36.73% and –6.93%, respectively for BB dam and –33.77% and –46.78%, respectively for
SK dam. While, the small percentage error of average reservoir inflows predicted by XGBoost univariate model varied
positively and negatively relative to observed values indicating both under–and overprediction.
Similar to XGBoost, the predictive results performed using LSTM (S2) for daily univariate prediction showed
the small discrepancy in minimum and average reservoir inflows of BB and SK dams. However, bigger discrepancy
was observed in maximum inflow prediction for these two dams. On the training datasets, discrepancy error in
minimum, average, and maximum reservoir inflows were +0.27 MCM, +0.26 MCM (+1.74% error), and –9.87 MCM
(–3.17% error), respectively, for BB dam, and +1.62 MCM, –0.05 MCM (–0.32% error), and –11.71 MCM (–5.46%),
respectively, for SK dam. On testing datasets, LSTM model exhibited small discrepancy error in minimum, average,
and maximum reservoir inflows by +0.27 MCM, +0.65 MCM (+6.00% error), and +18.05 MCM (+9.63% error),
respectively, for BB dam, and +1.14 MCM, +0.04 MCM (+0.30% error), and –11.14 MCM (–5.09%), respectively,
for SK dam.
For the daily multivariate prediction model using LSTM (S5), the discrepancy of minimum, average, and
maximum reservoir inflows of two dams were definitely close to those obtained from univariate prediction models
using two algorithms; XGBoost and LSTM. The daily minimum reservoir inflows performed by multivariate
prediction model for BB and SK dams were +0.20 MCM and +2.65 MCM, respectively on training datasets, and +0.21
MCM and +2.65 MCM, respectively on testing datasets. These predictive results were slightly overpredicted
compared to the minimum observed values. In addition, the daily average inflows for BB and SK dams on the training
datasets, were slightly underpredicted by –2.74% and –2.50%, respectively. On the testing datasets, overprediction
was observed, with values of +4.37% and +6.48%, respectively. Similar to the univariate models, multivariate
prediction models consistently underpredicted maximum reservoir inflows for both BB and SK dams, as obviously
found in both training and testing datasets.
All in all, the LSTM univariate model (S2) exhibited the smallest percentage errors in predicting minimum,
average, and maximum reservoir inflows for daily predictions compared to the XGBoost model (S1) and multivariate
prediction using LSTM (S5). Among all ML– and DL–based prediction models (S1, S2, and S5) for daily prediction,
underprediction of low reservoir inflows and overprediction of high reservoir inflows by both univariate and
multivariate prediction models were consistently emerged. However, LSTM–based individual prediction model is
recommended for high inflow prediction.





**Table 8** Percentage error of low, average and high reservoir inflow prediction

| Model Type | Optimal Prediction Model | | | | | Min. Reservoir Inflow (MCM) | | | Avg. Reservoir Inflow (MCM) | | | Max. Reservoir Inflow (MCM) | | |
|---|---|---|---|---|---|---|---|---|---|---|---|---|---|---|
| | Algorithm | Training: Testing | Learning Rate | Dam | Dataset | Obs. | Pred. | Δ (%) | Obs. | Pred. | Δ (%) | Obs. | Pred. | Δ (%) |
| S1: Daily Univariate Prediction | XGBoost | 80:20 | 0.1 | BB | Training | 0.00 | 0.17 | +0.17 | 17.52 | 16.71 | −0.81 (−4.62) | 311.46 | 197.05 | −114.41 (−36.73) |
| | | | | BB | Testing | 0.00 | 0.17 | +0.17 | 10.99 | 11.02 | +0.03 (+0.27) | 156.57 | 145.71 | −10.86 (−6.94) |
| | XGBoost | 80:20 | 0.1 | SK | Training | 0.00 | 3.03 | +3.03 | 17.32 | 16.70 | −0.62 (−3.58) | 221.87 | 146.95 | −74.92 (−33.77) |
| | | | | SK | Testing | 0.00 | 3.03 | +3.03 | 14.26 | 13.92 | −0.34 (−2.38) | 218.70 | 116.39 | −102.31 (−46.78) |
| S2: Daily Univariate Prediction | LSTM | 70:30 | 0.1 | BB | Training | 0.00 | 0.27 | +0.27 | 14.90 | 15.16 | +0.26 (+1.74) | 311.46 | 301.59 | −9.87 (−3.17) |
| | | | | BB | Testing | 0.00 | 0.27 | +0.27 | 10.83 | 11.48 | +0.65 (+6.00) | 187.34 | 205.39 | +18.05 (+9.63) |
| | LSTM | 70:30 | 0.1 | SK | Training | 0.00 | 1.62 | +1.62 | 15.81 | 15.76 | −0.05 (−0.32) | 214.42 | 202.71 | −11.71 (−5.46) |
| | | | | SK | Testing | 0.00 | 1.14 | +1.14 | 13.25 | 13.29 | +0.04 (+0.30) | 218.70 | 207.56 | −11.14 (−5.09) |
| S3: Monthly Univariate Prediction | XGBoost | 80:20 | 0.001 | BB | Training | 0.00 | 8.05 | +8.05 | 476.49 | 360.10 | −116.39 (−24.43) | 2,990.21 | 1,811.99 | −1,178.22 (−39.40) |
| | | | | BB | Testing | 12.99 | 10.87 | −2.12 (−16.32) | 370.31 | 359.75 | −10.56 (−2.85) | 2,373.51 | 1,740.76 | −632.75 (−26.66) |
| | | 70:30 | 0.001 | SK | Training | 61.48 | 78.04 | +16.56 (+26.94) | 543.94 | 479.97 | −63.97 (−11.76) | 3,095.97 | 2,076.67 | −1,019.30 (−32.92) |
| | | | | SK | Testing | 40.30 | 83.60 | +43.30 (+107.44) | 429.62 | 437.75 | +8.123 (+1.89) | 2,026.29 | 1,432.32 | −593.97 (−29.31) |
| S4: Monthly Univariate Prediction | LSTM | 70:30 | 0.1 | BB | Training | 0.00 | 56.53 | +56.53 | 460.61 | 479.25 | +18.64 (+4.05) | 2,877.23 | 2,373.81 | −503.42 (−17.50) |
| | | | | BB | Testing | 9.03 | 116.48 | +107.45 (+1,190) | 336.64 | 410.65 | +74.01 (+21.98) | 1,944.38 | 1,469.24 | −475.14 (−24.44) |
| | | 70:30 | 0.1 | SK | Training | 46.50 | 110.20 | +63.70 (+136.99) | 486.21 | 495.34 | +9.14 (+1.88) | 3,095.97 | 2,477.07 | −618.90 (−19.99) |
| | | | | SK | Testing | 40.30 | 112.55 | +72.25 (+179.28) | 410.14 | 460.03 | +49.89 (+12.17) | 2,026.29 | 1,845.41 | −180.88 (−8.93) |
| S5: Daily Multivariate Prediction | LSTM | 60:40 | 0.1 | BB | Training | 0.00 | −0.20 | −0.20 | 17.52 | 17.04 | −0.48 (−2.74) | 311.46 | 161.51 | −149.95 (−48.14) |
| | | | | BB | Testing | 0.00 | −0.21 | −0.21 | 10.99 | 11.47 | +0.48 (+4.37) | 156.57 | 151.09 | −5.48 (−3.50) |
| | | 60:40 | 0.1 | SK | Training | 0.00 | 2.65 | +2.65 | 18.38 | 17.92 | −0.46 (−2.50) | 221.87 | 121.05 | −100.82 (−45.44) |
| | | | | SK | Testing | 0.00 | 2.65 | +2.65 | 14.20 | 15.12 | +0.92 (+6.48) | 218.70 | 118.95 | −99.75 (−45.61) |
| S6: Monthly Multivariate Prediction | LSTM | 70:30 | 0.1 | BB | Training | 0.00 | 17.92 | +17.92 | 510.59 | 551.76 | +41.17 (+8.06) | 2,990.21 | 1,984.74 | −1,005.47 (−33.63) |
| | | | | BB | Testing | 1.57 | 16.09 | +14.52 (+924.84) | 323.19 | 455.97 | +132.78 (+41.08) | 2,373.51 | 1,481.92 | −891.59 (−37.56) |
| | | 60:40 | 0.1 | SK | Training | 61.48 | 38.49 | −22.99 (−37.39) | 563.63 | 551.53 | −12.097 (−2.15) | 3,095.97 | 2,258.06 | −837.91 (−27.06) |
| | | | | SK | Testing | 40.30 | 54.61 | +14.309 (+35.51) | 428.26 | 485.17 | +56.91 (+13.29) | 2,026.29 | 1,688.66 | −337.631 (−16.66) |


495    In view of the crucial role of accurate long–term reservoir inflow predictions in effective reservoir
management planning, this study focuses on developing robust monthly inflow prediction models. Even both
monthly univariate and multivariate prediction models efficiently learned and captured the inflow patterns, exhibiting
similarities to observed values, monthly prediction models (S3, S4, and S6) demonstrated larger deviations from
observed values with higher percentage errors compared to daily predictions (S1, S2, and S5). On the training
datasets of S3, a bigger discrepancy of monthly minimum reservoir inflows was +8.05 MCM and +16.56 MCM for
BB and SK dams, respectively. While, the monthly average and maximum reservoir inflows were −24.43% and
−39.40%, respectively for BB dam and −11.76% and −39.92%, respectively for SK dam. Base on the testing datasets,
the percentage errors particularly in monthly minimum and maximum inflows were found to be underpredicted by
−16.32% for BB dam and overpredicted by +107.44% for SK dam. The maximum reservoir inflows of both BB and
SK dams were underpredicted with the errors of −26.66% and −29.31%, respectively. Analysis of monthly minimum



and average inflows for both training and testing datasets of S4 and S6 revealed overprediction primarily for BB and
SK dams. Conversely, monthly maximum reservoir inflows were consistently underpredicted, exhibiting larger
percentage errors.
**4. Conclusion**
This study demonstrated the ability of ML–and DL–based univariate and multivariate prediction models to
predict daily and monthly reservoir inflows of BB and SK dams. Due to a wide range of successful applications of
ML and DL algorithms for hydrological prediction, XGBoost, a tree–based ensemble method, and LSTM, a deep
neural network, were selected for this study. To support real–time reservoir operation for the BB and SK dams, two
short–term prediction scenarios (S1 and S2) of daily univariate models and one scenario (S5) of daily multivariate
models were developed. In addition, two long–term prediction scenarios (S3 and S4) of monthly univariate models
and one scenario (S6) of monthly multivariate models were developed to support reservoir management planning. For
univariate prediction, the inflows of the BB and SK dams were predicted separately using two individual models. In
contrast, for multivariate prediction, a single model was developed to simultaneously predict the inflows of both the
BB and SK dams facilitating the integrated decision–making processes in the river basin. The results of all prediction
scenarios demonstrated that ML– and DL–based prediction models achieved higher statistical metrics evaluated in
terms of $R^2$, NSE, MSE, and RMSE in predicting daily reservoir inflow compared to monthly predictions. This is
because prediction modeling with ML and DL algorithms can handle and leverage larger datasets available in daily
prediction model, enabling them to learn and capture patterns more effectively. Based on a number of model
configuration experiments, individual XGBoost models mostly outperformed LSTM when tested on the datasets for
both daily and monthly univariate predictions. LSTM models consistently outperformed XGBoost when mostly
trained on larger datasets for both daily and monthly univariate and multivariate predictions. However, a slight
decrease in statistical metrics was apparently observed with smaller testing datasets. To enhance the robustness and
precision of LSTM–based forecasts, it is recommended to increase the testing dataset size during model validation
and employ cross-validation techniques to check for model overfitting. For input feature selection, the information on
past reservoir inflow including the previous time step and its moving average was considered as a significant predictor
to predict future inflow for all prediction scenarios. As LSTM can handle large datasets effectively, consequently,
rainfall and humidity data were also incorporated as additional input features indicating a substantial impact on
improved predictive performance. Ability to predict low, average, and high reservoir inflow by ML– and DL–based
prediction models were also assessed. Overall, LSTM–based univariate model distinctively exhibited the smallest
percentage errors in predicting minimum, average, and maximum reservoir inflows for daily and monthly predictions
compared to the XGBoost model and multivariate prediction using LSTM. Consequently, LSTM–based individual
daily and monthly prediction models are recommended for predicting low and high values of reservoir inflow during
critical events. In addition, monthly prediction models demonstrated larger discrepancy from observed values with
higher percentage errors compared to daily predictions. Among all ML– and DL–based prediction models for daily
and monthly predictions, underprediction of low reservoir inflows and overprediction of high reservoir inflows by
both univariate and multivariate prediction models were consistently existed. Therefore, extracting specific and
informative insights from the results of each prediction model and forecasting horizon can significantly enhance
decision–making support for both real–time reservoir operation and long–term reservoir management planning.
**Code and data availability statement**
The data and code are available upon request from the corresponding author.
**Interactive computing environment**
An R–based environment utilizing the Dplyr, ggplot2, imputeTS, Matrix, Metrics, RcppRoll, and Xgboost
libraries was used for developing XGBoost–based prediction models. A custom Python–based environment
leveraging RNN, Reticulate, Modeler, Keras, and other relevant Python libraries was employed for the development
of LSTM–based prediction models.
**Author contribution**
All authors contributed to the study conception and design. Material preparation, data collection and
analysis were performed by Areeya Rittima, Jidapa Kraisangka, and Pheeranat Dornpunya. The first draft of the
manuscript was written and reviewed by Areeya Rittima, Jidapa Kraisangka, Pheeranat Dornpunya and all authors
commented on previous versions of the manuscript. All authors read and approved the final manuscript.



**Completing interests**

The authors declare that they have no known competing financial interests or personal relationships that could have appeared to influence the work reported in this paper.

The authors declare that they have no conflict of interest.

**Acknowledgements**

This research was funded by the Office of the National Research Council of Thailand (NRCT). The authors are grateful to the Royal Irrigation Department (RID), Electricity Generating Authority of Thailand (EGAT) for providing research data.

**Financial support**

This work was financially supported by the National Research Council of Thailand (NRCT), grant numbers SIP6230022. A Rittima received research support from NRCT in 2021.

**Review statement**

This study demonstrated the predictability of Machine Learning (ML)– and Deep Learning (DL)–based univariate and multivariate predictions of reservoir inflows of Bhumibol (BB) and Sirikit (SK), two major dams in the Chao Phraya River Basin. XGBoost, tree–based ensemble–, and LSTM, deep neural network–based algorithms were selected for development of daily and monthly prediction models. Input features selection and configuration design of two different types of daily and monthly reservoir inflow prediction models; (1) univariate prediction with XGBoost and LSTM algorithms and (2) multivariate prediction with LSTM algorithm, were definitely highlighted. In the last step, predictability of predicting low and high reservoir inflow values of these models were accordingly explored to assess their statistical performances.

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
