# Peer review of "Predictive Performances of Machine Learning– and Deep Learning–Based Univariate and Multivariate 2 Reservoir Inflow Predictions in the Chao Phraya River Basin"

_EGUsphere, 2025_

## Author Comment (AC1)

**Reply on RC2_HESS Journal**

| Reviewer's comment | Reply |
|---|---|
| The paper investigates the ability of Machine Learning (XGBoost) and Deep Learning (LSTM) models to predict daily and monthly reservoir inflows for the Bhumibol (BB) and Sirikit (SK) dams in Thailand's Chao Phraya River Basin. The authors evaluate both univariate models (predicting each dam separately) and multivariate models (predicting both dams simultaneously). | Thank you so much for your insightful and helpful feedback on our manuscript. We appreciate your time scarification to carefully review our research work and provide detailed comments. Absolutely, our research aims to propose applications of AI algorithms for both univariate prediction, where reservoir inflow of two major dams in the same river basin is predicted separately, and multivariate prediction, where reservoir inflow of two dams are predicted simultaneously in one model. |
| The multivariate approach is tested only with LSTM and not with XGBoost. Why do the authors not apply a multivariate approach using XGBoost as well? | Thank you for your suggestion regarding the approach. Our primary focus was to explore the multivariate modeling capability of LSTM, as it is designed to handle complex sequential datasets and capture long–term time dependencies through its memory cells and gates. Compared to XGBoost, we experienced that LSTM is more precise for modeling complex temporal relationships among multiple features in a multivariate setting. |
| The paper is generally well structured and presents a clear methodology. The correlation analysis presented in Section 2.1 serves as a useful preliminary step for selecting relevant input features. However, this approach would benefit from being complemented by a more in–depth evaluation of feature importance based on the model's actual behavior. This is particularly relevant for complex architectures such as LSTM, where the relationship between inputs and outputs is not always easily. | We appreciate your feedback on in–depth evaluation of feature importance. We will address this point and add discussion carefully in our response and revisions. |
| Have the authors employed a systematic cross–validation strategy? In Section 2.3, the XGBoost configuration specifies only a 2–fold validation, which is the bare minimum. It would be more appropriate to use 5– or 10–fold cross–validation to ensure more reliable results. Furthermore, for the LSTM models, it appears that no form of cross–validation has been applied. To strengthen the robustness of the comparative analysis, a more comprehensive approach–such as k–fold crossvalidation–should be adopted and clearly described for all models. | We appreciate the reviewer's insightful feedback regarding the k–fold cross–validation strategy. We agree that a systematic approach, such as k–fold cross–validation, would enhance the robustness of the comparative analysis. However, we employed multiple train–test splits to ensure evaluation across different data partitions. This limitation will acknowledge in the discussion section of the paper. |
| The description of the algorithms, while thorough, is excessively detailed and could be streamlined by referencing existing literature where appropriate. Additionally, the paper lacks a critical discussion of the limitations inherent to the algorithms used. The conclusion section would also be strengthened by including suggestions | We appreciate your insightful comments regarding the descriptions of the algorithm selected, the critical discussion of limitations, and the conclusion section. We have realized the importance of background and theory for each AI algorithm used in our study, and have therefore already |

| Reviewer's comment | Reply |
|---|---|
| for future research directions and potential areas of improvement. | systematically structured, organized and presented the descriptions of algorithms and equations in our manuscript. To further enhance this, we will incorporate more references to existing literature and add a comprehensive discussion on the limitations of the algorithms used.
We will strengthen the conclusion section by suggesting future research directions and potential areas of improvement. We will ensure that your valuable comments are adequately addressed in the revised manuscript. |
| In Table 6, the test metrics unexpectedly outperform the training metrics–for example, in the XGBoost monthly model (S3–BB), the NSE increases from 0.411 (training) to 0.675 (testing). Generally, this should not occur, as the model is specifically trained on the training data, while the test set is meant to evaluate its ability to generalize to unseen data. Although it is technically possible for test metrics to exceed training metrics, such results often indicate potential issues–
such as noise or outliers in the training set, an under–trained model, or a methodological flaw in the data splitting process, especially if temporal ordering is not preserved. | We appreciate the reviewer's insights. The unexpected cases where the test metrics outperform the training metrics may indicate potential overfitting of the XGBoost model or limitations in its ability to generalize effectively across different data partitions. In contrast, the LSTM model, which is specifically designed for sequential data and temporal dependencies, demonstrates more consistent performance across various data splits. We view this as a potential advantage of LSTM over XGBoost in this context and will discuss this limitation in the discussion section. |
| It would be important to clarify whether the authors used a random shuffling approach to split the dataset into training and testing sets. | Thank you for the clarification. We did not apply random shuffling when splitting the dataset into training and testing sets. This decision was made to preserve the temporal dependencies in the data, ensuring that the model is trained and evaluated in a manner consistent with the nature of sequential time–series forecasting. |
| Moreover, Table 6 shows that models using daily resolution consistently outperform those using monthly resolution across all evaluation metrics. Intuitively, one might expect the opposite–that monthly predictions would perform better–as temporal aggregation typically reduces noise and smooths out short–term variability. However, this counterintuitive outcome warrants further investigation. The authors should explore this aspect in more depth, discussing possible reasons. | We appreciate your insightful comment on the predictive performance between daily prediction model and monthly prediction model. Absolutely, monthly prediction is expected to perform better due to the reduced noise and smoothen variability of monthly values. However, we have found that shorter length of monthly sequential dataset significantly limits ability to train and test to closely match with the observed monthly data especially when LSTM algorithm were applied. However, we will ensure you that this point will be included in the discussion section of our revised manuscript. |
| In Section 3.2, comparing models on different test sets may compromise the reliability of the evaluation. To ensure a fair and consistent comparison, it is recommended to exclude a predefined period of observations and use this subset as a shared validation set for all models. This approach guarantees that all models are assessed on the same data, minimizing variability and potential biases in | We understand your point regarding a fair and consistent comparison among different scenarios of prediction models. However, we are unclear about your point regarding excluding a predefined period of observations and using the subset as a shared validation set for all models. |

| Reviewer's comment | Reply |
|---|---|
| performance comparison. | As you know, our experiments were designed based on the proposed input features and model types. Consequently, the evaluation of model predictability and comparison was based on these scenarios. In Section 3.2, we highlight the optimal model configuration for each prediction scenario across all experiment runs by altering hyperparameter settings. For example, in S1: XGBoost for BB, there are three experiments (dBB–01, dBB–02, and dBB–03, referred to as submodels). For these, predefined periods of the dataset for model training and testing were set as the same (an 80:20 split). In other words, a consistent comparison was made among the designed experiments of each prediction scenario. However, we will add this point to the recommendations section and include a clarifying footnote in Table 7 in the revised manuscript. |
| In Table 8, the minimum and average observed values (i.e., those labeled as "Obs.") should be identical across all models tested on the same dataset, unless there were differences in data splitting or preprocessing. These "Obs." values represent the actual ground truth measurements used to evaluate model performance. Therefore, if the training set remains unchanged across models, the minimum, average, and maximum observed values in the training set should be the same. The same applies to the test set: if the data used is consistent, the observed statistics must also be consistent. However, there are clear inconsistencies in Table 8. For example, in the SK monthly models (S3 vs. S4), the minimum observed inflow in the training set changes from 61.48 MCM (S3) to 46.50 MCM (S4). | We appreciate your detailed feedback on the inconsistency of observed data presented in Table 8. We agree with you, the basic statistics of observed data (min., avg. and max.) should be consistent among relevant prediction scenarios (like S3 and S4). We will check training and testing ratios used and update all values in the revised manuscript. Thank you so much. |
| The Authors acknowledge the tendency of the models to underestimate low flows and overestimate high flows (Table 8). What do the authors propose to address the consistent overestimation of maximum values and underestimation of minimum values? Do they suggest any strategies or methodologies to mitigate this issue? | We have found our mistake of concluding remarks within the abstract section. Based on the discrepancy percentages presented in Table 8, low flow predictions are likely overestimated (positive percentage values), while high flow predictions are underestimated (negative percentage values). Addressing this overestimation of low flow and underestimation of high flow is crucial and informative for real–time reservoir operation, particularly for mitigating flood and drought risks and for operational practice by dam operators and decision makers. To incorporate these issues in our study, we conducted the evaluation of predictability of ML and DL – based prediction models to predict low and high flow in the Section 3.3. The percentage discrepancies are presented. |

| Reviewer's comment | Reply |
|---|---|
| | Furthermore, we will add recommendations for relevant strategies and methodologies to the conclusion section. |
| | Specifically, we will suggest: |
| | (1) The design of advanced loss functions for XGBoost and LSTM algorithms to address the observed predictive biases. |
| | (2) Application of data preprocessing & feature engineering techniques for extreme flow conditions (low and high flow). For example, target transformation techniques like logarithm or power transformation can be applied to compress big values and expand small values to help better capture wide range of flow behavior. |
| Minor comments:
 • Improve the resolution of the images; in their current state, they are very difficult to read.
 • The captions of the tables and figures should be more descriptive of the content they present.
 • Figure 7: The images A.1, A.2, B.1, and B.2 do not use the same scale, which makes direct comparison difficult. | Thank you so much for your detailed feedback on the graphical design of all images in the manuscript and their captions. We will improve their resolution of the images for visibility and clarity and adjust the scale to facilitate easy comparison and effective communication. |

---

## Author Comment (AC2)

**Reply on RC1_HESS Journal**

| Reviewer's comment | Reply |
|---|---|
| This manuscript explores the application of two widely known data-driven algorithms—XGBoost and LSTM—in both univariate and multivariate modes for daily and monthly inflow predictions at two key reservoirs in the Chao Phraya River Basin. The topic is timely and relevant in the context of AI-driven hydrological forecasting. However, the manuscript, in its current form, fails to meet the scientific standards and novelty threshold expected by Hydrology and Earth System Sciences. The work is largely confirmatory, methodologically simplistic, and lacks both theoretical depth and critical interpretation. It represents an incremental application of well-established techniques without significant advancement in methodology, theory, or hydrological insight. Below are my detailed comments: | Thank you very much for your insightful and helpful feedback on our manuscript. We appreciate your time scarification to carefully review our research work and provide detailed comments.

We acknowledge the points raised regarding the scientific standards and novelty of our manuscript for Hydrology and Earth System Sciences. However, our study aimed to adopt modern AI-based techniques and integrate them with stochastic modeling concepts for hydrologic time series outlined in Salas, J.D., Delleur, J.W., Yevjevich, V., Lane, W.L., 1980. Applied Modelling of Hydrologic Time Series:Univariate And Multivariate Prediction, detailed in Chapter7: Multivariate Modelling of Hydrologic Time Series. We believe that the development of a hybrid approach that leverages the strengths of both data-driven AI incorporated with the inherent understanding of temporal dependencies in stochastic time series models offers novelty and valuable predictive option within this research area.

Furthermore, recognizing that water resource systems with multiple reservoirs are often managed on a watershed basis, developing individual models for each reservoir's inflow in a complex system may not effectively support reservoir operators, particularly during critical events for rapid decision making. Therefore, our study proposes constructing a single prediction model to predict two outputs in the same watershed system by leveraging the capabilities of the LSTM algorithm.

We understand your point that the application of XGBoost and LSTM algorithms in this context might appear largely confirmatory. Importantly, our intention is to share the concept of developing single model for multivariate prediction and compare its performance with the conventionally univariate model by adopting AI algorithms in a specific basin to explore the practical applicability and its predictive performances for watershed-based management support. However, we recognize that the current presentation may not adequately highlight specific challenges we encountered in this application. |

| Reviewer's comment | Reply |
|---|---|
| | We appreciate your feedback on the methodological simplicity. However, a key focus of our research is the integration of AI-driven techniques with a stochastic-based model for hydrological forecasting. Our selection of input features was carefully considered based on the physically-hydrological system and long-term time series data, aiming to capture the relevant hydrological phenomena in the basin. We believe this specific integration offers novelty in the area of hydrological prediction. For our revision or future work, we will certainly consider exploring more complex related methodologies and well-demonstrated algorithms.

We recognize the importance of the theoretical depth and critical interpretation of our findings. For our revision, we will revisit and explore more on the existing literature and our results to provide a more in-depth discussion of the hydrological processes and the implications of our model predictive performance.

We appreciate your feedback regarding insignificant advancement in methodology, theory, or hydrological insight.
We are committed to explore more and to provide more profound hydrological insights based on our findings.

Thank you again for your constructive criticism on our work. We are committed to revising the manuscript to meet the expectations of Hydrology and Earth System Sciences later. |
| 1. Despite the claim of contributing to reservoir inflow forecasting through multivariate models, the study does not introduce any methodological innovation. The application of XGBoost and LSTM, both extensively used in hydrology, adds no novelty unless combined with a new model architecture, uncertainty treatment, explainability component, or integration with process-based models. The experimental setting is rudimentary, and the results primarily confirm what has already been established in dozens of prior studies. Moreover, the assertion that multivariate prediction of inflows has rarely been studied is not substantiated and contradicts recent literature. The references cited are selective and outdated, omitting more advanced hybrid or physics-informed ML approaches currently under development in the hydrological community. | We appreciate your detailed and critical feedback on our manuscript. We acknowledge your concerns regarding the methodological innovation and novelty of our work, as well as the experimental setup and literature review. However, we will address each of these points carefully in our response and revisions. |
| 2. The manuscript fails to clearly define its scientific objectives or hypotheses. The rationale behind | We appreciate your feedback highlighting the lack of clearly defined scientific objectives, |

| Reviewer's comment | Reply |
|---|---|
| comparing univariate and multivariate approaches is weakly stated and not embedded in a theoretical or operational framework. The problem formulation is generic and reads more like a technical report than a scientific investigation. | hypotheses, and a strong rationale within a theoretical or operational framework. For our revision, we will clearly define scientific objectives, explicitly state the hypotheses and robust rational of our works. These focused efforts will be made to enhance the quality and scientific investigation of our work. |
| 3. The literature review is overly descriptive and lacks synthesis. It resembles an annotated bibliography rather than a critical narrative. Foundational works on multivariate time series modeling, ensemble learning, recent benchmarks on hybrid models, and the emerging field of physics-informed ML in hydrology are all missing. Furthermore, no discussion is provided on model explainability, uncertainty quantification, or generalization capacity, all of which are central themes in the current hydrological ML research agenda. | We appreciate your detailed and critical feedback on our literature review. we will thoroughly revise to develop robust literature review and address these shortcomings comprehensively following your valuable feedback. |
| 4. The methodology exhibits some critical flaws:
    - No hyperparameter optimization strategy is described beyond brute-force listing of combinations.
    - Feature selection is based solely on Pearson correlation, ignoring non-linear dependencies or mutual information approaches.
    - The study does not address overfitting or generalization. Despite LSTM being known for susceptibility to overfitting, no regularization, dropout, or model selection strategy is employed.
    - No benchmark model is used for reference, which is standard in HESS-level contributions. | We appreciate your critical feedback highlighting several important flaws in our methodology. We acknowledge the shortcomings regarding hyperparameter optimization, feature selection, the lack of measures to address overfitting and generalization, and the absence of a benchmark model. We recognize the significance of these issues and will thoroughly revise our methodology to strengthen our work. |
| 5. The manuscript presents no discussion of data quality, treatment of missing values, stationarity, or outlier detection. | We appreciate your feedback pointing out the absence of any discussion regarding data quality, treatment of missing values, stationarity, or outlier detection in our manuscript. We acknowledge that these are critical aspects of hydrological data analysis and modeling, and their omission is a significant oversight. We will thoroughly address these points in our revised manuscript.

In particular, we will add a section of data quality assessment covering data quality, treatment of missing values, stationarity analysis, and outlier detection. |
| 6. The results section is overly descriptive, listing metrics without proper analysis or critical discussion. Additionally, the model performances reported are relatively modest, especially for monthly inflow prediction, yet are uncritically presented as acceptable. | We are thankful for your feedback pointing out that our results section is overly descriptive, lacking proper analysis and critical discussion, and that the relatively modest model performances, particularly for monthly inflow prediction, were presented uncritically. We acknowledge these valid concerns and will |

| Reviewer's comment | Reply |
|---|---|
| | thoroughly revise this section to address them carefully in our revision. |
| 7. The discussion does not provide new hydrological or methodological insight. There is no exploration of why certain models perform better under given conditions, nor any effort to relate findings to hydrological processes. The difference in performance between the two dams, for instance, is acknowledged but not explained. | We appreciate your feedback pointing out the lack of new hydrological or methodological insight in our discussion, the absence of exploration into why certain models perform better under specific conditions, and the lack of connection to underlying hydrological processes. We also acknowledge that the performance differences between the two dams were not adequately explained. For our revision, we will strengthen these as significant limitations and will address them carefully. |
| 8. The implications for operational decision-making—often emphasized in the introduction—are not convincingly revisited. | Thank you so much for your insightful feedback. Since the implications for operational decision-making, which we emphasized in the introduction, were not convincingly revisited in the later sections of the manuscript, particularly the discussion and conclusion. So, we acknowledge this oversight and will address it directly in our revision. |
| 9. The conclusions are largely a restatement of the results, without any critical reflection or forward-looking perspective. The authors do not acknowledge the substantial limitations of their study—particularly the lack of generalization, interpretability, and robustness of the models. | We appreciate your feedback pointing out that our conclusions largely restate the results and lack critical reflection, a forward-looking perspective, an acknowledgment of the study's limitations, and robustness of the models. We recognize the importance of a more robust and insightful conclusion and will thoroughly revise this section corresponding to your detailed comments. |
| 10. The manuscript suffers from structural repetition and verbosity. Some figures (e.g., radar plots) are poorly designed and do not enhance interpretability. | We appreciate your feedback regarding the structural repetition, verbosity, and the design of some figures. We acknowledge these shortcomings and will address them carefully in our revision. For the structural repetition and verbosity: we will thoroughly review the manuscript to erase any instances of unnecessary repetition. For figure design (e.g., radar plots): some figures, such as the radar plots, may not be effectively enhancing interpretability. We will redesign these figures with a focus on clarity and visual communication later. |